# Demystifying Softmax Gating Function in Gaussian Mixture of Experts

**Huy Nguyen**[†]    **TrungTin Nguyen**[◇]    **Nhat Ho**[†]

Department of Statistics and Data Sciences, The University of Texas at Austin[†]

Univ. Grenoble Alpes, Inria, CNRS, Grenoble INP, LJK[◇]

{huynm, minhnhat}@utexas.edu, trung-tin.nguyen@inria.fr

## Abstract

Understanding the parameter estimation of softmax gating Gaussian mixture of experts has remained a long-standing open problem in the literature. It is mainly due to three fundamental theoretical challenges associated with the softmax gating function: (i) the identifiability only up to the translation of parameters; (ii) the intrinsic interaction via partial differential equations between the softmax gating and the expert functions in the Gaussian density; (iii) the complex dependence between the numerator and denominator of the conditional density of softmax gating Gaussian mixture of experts. We resolve these challenges by proposing novel Voronoi loss functions among parameters and establishing the convergence rates of maximum likelihood estimator (MLE) for solving parameter estimation in these models. When the true number of experts is unknown and over-specified, our findings show a connection between the convergence rate of the MLE and a solvability problem of a system of polynomial equations.

## 1   Introduction

Softmax gating Gaussian mixture of experts [32, 37], a class of statistical machine learning models that combine multiple simpler models, known as expert functions of the covariates, via softmax gating networks to form more complex and accurate models, has found widespread use in various applications, including speech recognition [51, 64, 65], natural language processing [14, 20, 17, 54, 21], computer vision [52, 3, 15, 44], and other applications [26, 49, 7, 8, 48, 5, 6]. Regarding the applications of the softmax gating Gaussian mixture of experts in medicine [43] and physical sciences [39], the parameters of each expert function play an important role in capturing the heterogeneity of data. Thus, the main objective of these works is to conduct statistical inference for those parameters, which leads to a need for convergence rates of parameter estimation in the softmax gating Gaussian mixture of experts. However, a comprehensive theoretical understanding of parameter estimation in that model has still remained a long-standing open problem in the literature.

Parameter estimation has been studied quite extensively in standard mixture models. In his seminal work, Chen et al. [9] established the convergence rate $\mathcal{O}(n^{-1/4})$ of parameter estimation in over-fitted univariate mixture models, namely, the settings when the number of true components is unknown and over-specified, and the family of distributions is strongly identifiable in the second order, e.g., location Gaussian distributions. That slow and non-standard rate is due to the collapse of some parameters into single pararameter or the convergence of weights to zero, which leads to the singularity of Fisher information matrix around the true parameters. Then, Nguyen et al. [50] and Ho et al. [29] utilized Wasserstein metrics to achieve this rate under the multivariate settings of second-order strongly identifiable mixture models. Recently, Ho et al. [28] demonstrated that rates of the MLE can strictly depend on the number of over-specified components when the mixture models are not strongly

37th Conference on Neural Information Processing Systems (NeurIPS 2023).

identifiable, such as location-scale Gaussian mixtures. The minimax optimal behaviors of parameter estimation were studied in [27, 46]. From the computational side, the statistical guarantee of the expectation-maximization (EM), e.g., [12], and moment methods had also been studied under both exact-fitted [2, 1, 25] and over-fitted settings [19, 18, 61, 16, 62] of mixture models.

Compared to mixture models, there has been less research on parameter estimation of mixture of experts. When the gating networks are independent of the covariates, Ho et al. [30] employed the generalized Wasserstein loss function [59] to study the convergence rates of parameter estimation in Gaussian mixture of experts. They proved that these rates are determined by the algebraic independence of the expert functions and the partial differential equations with respect to the parameters. Later, Do et al. [13] extended these results to general mixture of experts with covariate-free gating network. Statistical guarantees of optimization methods for solving parameter estimation in Gaussian mixture of experts with covariate-free gating functions were studied in [11, 67, 41, 63]. When the gating networks are softmax functions, parameter estimation becomes more challenging to understand due to the complex structures of the softmax gating function in the Gaussian mixture of experts. Before describing these phenomena in further details, we begin by formally introducing the softmax gating Gaussian mixture of experts and related notions.

**Problem setting:** Assume that $(X_1, Y_1), \ldots, (X_n, Y_n) \in \mathbb{R}^d \times \mathbb{R}$ are i.i.d. samples drawn from the softmax gating Gaussian mixture of experts of order $k_*$ whose conditional density function $g_{G_*}(Y|X)$ is given by:

$$g_{G_*}(Y|X) := \sum_{i=1}^{k_*} \frac{\exp((\beta_{1i}^*)^\top X + \beta_{0i}^*)}{\sum_{j=1}^{k_*} \exp((\beta_{1j}^*)^\top X + \beta_{0j}^*)} \cdot f(Y|(a_i^*)^\top X + b_i^*, \sigma_i^*), \tag{1}$$

where $f(.|\mu, \sigma)$ is a Gaussian density function with mean $\mu$ and variance $\sigma$. Here, we define $G_* := \sum_{i=1}^{k_*} \exp(\beta_{0i}^*) \delta_{(\beta_{1i}^*, a_i^*, b_i^*, \sigma_i^*)}$ as a true but unknown *mixing measure*, that is, a combination of Dirac measures $\delta$ associated with true parameters $\theta_i^* := (\beta_{0i}^*, \beta_{1i}^*, a_i^*, b_i^*, \sigma_i^*)$. Notably, $G_*$ is not necessarily a probability measure as the summation of its weights can be different from one. For the purpose of the theory, we assume that $\theta_i^* \in \Theta \subset \mathbb{R} \times \mathbb{R}^d \times \mathbb{R}^d \times \mathbb{R} \times \mathbb{R}_+$ where $\Theta$ is a compact set, and $X \in \mathcal{X} \subset \mathbb{R}^d$ where $\mathcal{X}$ is a bounded set. Furthermore, we let $(a_1^*, b_1^*, \sigma_1^*), \ldots, (a_{k_*}^*, b_{k_*}^*, \sigma_{k_*}^*)$ be pairwise distinct and at least one among $\beta_{11}^*, \ldots, \beta_{1k_*}^*$ be non-zero to guarantee the dependence of softmax gating function on the covariate $X$. Finally, we assume that the covariate $X$ follows a continuous distribution to ensure that the softmax gating Gaussian mixture of experts is at least identifiable up to translations (see Proposition 1).

**Maximum likelihood estimation.** Since the value of true order $k_*$ is unknown in practice, to estimate the unknown parameters in the softmax gating Gaussian mixture of experts (1), we consider using maximum likelihood estimation (MLE) within a class of all mixing measures with at most $k$ components, which is defined as follows:

$$\widehat{G}_n \in \underset{G \in \mathcal{O}_k(\Theta)}{\arg\max} \frac{1}{n} \sum_{i=1}^{n} \log(g_G(Y_i|X_i)), \tag{2}$$

where $\mathcal{O}_k(\Theta) := \{G = \sum_{i=1}^{k'} \exp(\beta_{0i}) \delta_{(\beta_{1i}, a_i, b_i, \sigma_i)} : 1 \le k' \le k \text{ and } (\beta_{0i}, \beta_{1i}, a_i, b_i, \sigma_i) \in \Theta\}$. To guarantee that the MLE $\widehat{G}_n$ is a consistent estimator of $G_*$, we need $k \ge k_*$. In this paper, we study the convergence rate of the MLE $\widehat{G}_n$ to the true mixing measure $G_*$ under both the *exact-fitted* settings, namely when $k = k_*$, and the *over-fitted* settings, namely when $k > k_*$, of the softmax gating Gaussian mixture of experts.

**Fundamental challenges from the softmax gating function:** There are three fundamental challenges arising from the softmax gating function that create various obstacles in our convergence analysis:

**(i)** Firstly, parameters $\beta_{1i}^*, \beta_{0i}^*$ of the softmax gating function are not identifiable as those of the covariate-independent gating function in previous work. Instead, they are identifiable up to translations, that is, the softmax gating value does not change when we translate $\beta_{0i}^*$ to $\beta_{0i}^* + t_1$ and $\beta_{1i}^*$ to $\beta_{1i}^* + t_2$ for any $t_1 \in \mathbb{R}$ and $t_2 \in \mathbb{R}^d$. As a consequence, we need to introduce an infimum operator in the Voronoi loss functions (see equations (4) and (6)) to deal with this issue.

**(ii)** Secondly, a key step in our proof techniques is to decompose the density discrepancy $g_{\widehat{G}_n}(Y|X) - g_{G_*}(Y|X)$ into a linear combination of linearly independent elements using Taylor expansions.

However, since the numerators and denominators of softmax gating functions are dependent, we cannot apply the Taylor expansions directly to that density discrepancy as in previous work [30, 13]. Moreover, there are two intrinsic interactions between parameters of the softmax gating's numerators and the Gaussian density function via the following partial differential equations (PDEs), which induce a lot of linearly dependent derivative terms in the Taylor expansions:

$$\frac{\partial^2 u}{\partial \beta_1 \partial b} = \frac{\partial u}{\partial a}; \qquad \frac{\partial^2 u}{\partial b^2} = 2\frac{\partial u}{\partial \sigma}, \tag{3}$$

where $u(Y|X; \beta_1, a, b, \sigma) := \exp(\beta_1^\top X) \cdot f(Y|a^\top X + b, \sigma)$. Therefore, it takes us great effort to group those linearly dependent terms together to obtain the desired linear combination of linearly independent terms.

**(iii)** Lastly, given the above linear combination of linearly independent elements, when the density estimation $g_{\widehat{G}_n}(Y|X)$ converges to the true density $g_{G_*}(Y|X)$, the associated coefficients in that combination also go to zero. Then, via some transformations, those limits lead to a system of polynomial equations introduced in equation (9). This system admits a much more complex structure than what considered in previous work [30, 13].

These fundamental challenges from the softmax gating function suggest that the previous loss functions, such as Wasserstein distance [50, 28, 30], being employed to study parameter estimation in standard mixture models or mixture of experts with covariate-free gating functions are no longer sufficient as they heavily rely on the assumptions that the weights of these models are independent of the covariates.

**Main contributions:** To tackle these challenges of the softmax gating function, we propose two novel Voronoi losses among parameters and establish the lower bounds of the Hellinger distance, denoted as $h(\cdot, \cdot)$, of the mixing densities of softmax gating Gaussian mixture of experts in terms of these Voronoi losses to capture the behaviors of the MLE. Our results can be summarized as follows (see also Table 1):

**1. Exact-fitted settings**: When $k = k_*$, we demonstrate that the Hellinger lower bound $\mathbb{E}_X[h(g_G(\cdot|X), g_{G_*}(\cdot|X))] \geq C \cdot \mathcal{D}_1(G, G_*)$ holds for any mixing measure $G \in \mathcal{O}_k(\Theta)$, where $C$ is some universal constant and the Voronoi metric $\mathcal{D}_1(G, G_*)$ is defined as:

$$\mathcal{D}_1(G, G_*) := \inf_{t_1, t_2} \sum_{j=1}^{k_*} \left[ \sum_{i \in \mathcal{A}_j} \exp(\beta_{0i}) \|(\Delta_{t_2}\beta_{1ij}, \Delta a_{ij}, \Delta b_{ij}, \Delta \sigma_{ij})\| \right.$$

$$\left. + \left| \sum_{i \in \mathcal{A}_j} \exp(\beta_{0i}) - \exp(\beta_{0j}^* + t_1) \right| \right], \tag{4}$$

where $\Delta_{t_2}\beta_{1ij} := \beta_{1i} - \beta_{1j}^* - t_2$, $\Delta a_{ij} := a_i - a_j^*$, $\Delta b_{ij} := b_i - b_j^*$, $\Delta \sigma_{ij} := \sigma_i - \sigma_j^*$. The infimum over $t_1 \in \mathbb{R}$ and $t_2 \in \mathbb{R}^d$ is to account for the identifiability up to the translation of $(\beta_{0j}^*, \beta_{1j}^*)_{j=1}^{k_*}$. Furthermore, $\mathcal{A}_j$ is a Voronoi cell of mixing measure $G$ generated by the true component $\omega_j^* := (\beta_{1j}^*, a_j^*, b_j^*, \sigma_j^*)$ for all $1 \leq j \leq k_*$ [47], which is defined as follows:

$$\mathcal{A}_j \equiv \mathcal{A}_j(G) := \{i \in \{1, 2, \ldots, k\} : \|\omega_i - \omega_j^*\| \leq \|\omega_i - \omega_\ell^*\|, \forall \ell \neq j\}, \tag{5}$$

where we denote $\omega_i := (\beta_{1i}, a_i, b_i, \sigma_i)$. It is worth noting that the cardinality of each Voronoi cell $\mathcal{A}_j$ indicates the number of components of $G$ approximating the true component $\omega_j^*$ of $G_*$. As $\mathbb{E}_X[h(g_{\widehat{G}_n}(\cdot|X), g_{G_*}(\cdot|X))] = \mathcal{O}(n^{-1/2})$, that lower bound of Hellinger distance indicates that $\mathcal{D}_1(\widehat{G}_n, G_*) = \mathcal{O}(n^{-1/2})$. Therefore, the rates of estimating $\exp(\beta_{0j}^*), \beta_{1j}^*$ (up to translations) and $a_j^*, \beta_j^*, \sigma_j^*$ are of optimal order $\mathcal{O}(n^{-1/2})$.

**2. Over-fitted settings**: When $k > k_*$, the lower bound of Hellinger distance in terms of the Voronoi metric $\mathcal{D}_1$ in the exact-fitted settings is no longer sufficient due to the collapse of softmax of vectors in possibly $k$ dimensions to softmax of vectors in $k_*$ dimensions. Our approach is to define more

fine-grained Voronoi metric $\mathcal{D}_2(G, G_*)$ to capture such collapse, which is given by:

$$\mathcal{D}_2(G, G_*) := \inf_{t_1, t_2} \sum_{j:|\mathcal{A}_j|>1} \sum_{i \in \mathcal{A}_j} \exp(\beta_{0i})\Big(\|(\Delta_{t_2}\beta_{1ij}, \Delta b_{ij})\|^{\bar{r}(|\mathcal{A}_j|)} + \|(\Delta a_{ij}, \Delta\sigma_{ij})\|^{\bar{r}(|\mathcal{A}_j|)/2}\Big)$$

$$+ \sum_{j:|\mathcal{A}_j|=1} \sum_{i \in \mathcal{A}_j} \exp(\beta_{0i})\|(\Delta_{t_2}\beta_{1ij}, \Delta a_{ij}, \Delta b_{ij}, \Delta\sigma_{ij})\| + \sum_{j=1}^{k_*} \Big|\sum_{i \in \mathcal{A}_j} \exp(\beta_{0i}) - \exp(\beta_{0j}^* + t_1)\Big|, \quad (6)$$

for any mixing measure $G \in \mathcal{O}_k(\Theta)$. Here, the values of function $\bar{r}(\cdot)$ are determined by the solvability of a system of polynomial equations defined in equation (9). We then show in Lemma 1 that $\bar{r}(2) = 4$, $\bar{r}(3) = 6$, and we conjecture that $\bar{r}(m) = 2m$ for any $m \geq 2$.

In high level, the aforementioned system of polynomial equations arises from the PDEs in equation (3) when we establish the lower bound $\mathbb{E}_X[h(g_G(\cdot|X), g_{G_*}(\cdot|X))] \geq C' \mathcal{D}_2(G, G_*)$ for any $G \in \mathcal{O}_k(\Theta)$ for some universal constant $C'$. Since $\mathbb{E}_X[h(g_{\widehat{G}_n}(\cdot|X), g_{G_*}(\cdot|X))] = \mathcal{O}(n^{-1/2})$, we also have $\mathcal{D}_2(\widehat{G}_n, G_*) = \mathcal{O}(n^{-1/2})$ under the over-fitted settings of the softmax gating Gaussian mixture of experts. As a consequence, the rates for estimating true parameters whose Voronoi cells have only one component of the MLE are of order $\mathcal{O}(n^{-1/2})$. On the other hand, for true parameters $\exp(\beta_{0j}^*), \beta_{1j}^*, a_j^*, b_j^*, \sigma_j^*$ whose Voronoi cells have more than one component of the MLE, the estimation rates are respectively $\mathcal{O}(n^{-1/2\bar{r}(|\mathcal{A}_j|)})$ for $\beta_{1j}^*, b_j^*$, $\mathcal{O}(n^{-1/\bar{r}(|\mathcal{A}_j|)})$ for $a_j^*, \sigma_j^*$, and $\mathcal{O}(n^{-1/2})$ for $\exp(\beta_{0j}^*)$. This rich spectrum of parameter estimation rates is due to the complex interaction between the softmax gating and the expert functions.

Table 1: Summary of density estimation and parameter estimation rates in the softmax gating Gaussian mixture of experts under both the exact-fitted and over-fitted settings. Recall that the cardinality of each Voronoi cell $\mathcal{A}_j$ gives the number of fitted components approximating true component $\omega_j^* = (\beta_{1j}^*, a_j^*, b_j^*, \sigma_j^*)$ (see equation (5)). Furthermore, the notation $\bar{r}(|\mathcal{A}_j|)$ stands for the solvability of the system of polynomial equations (9). For instance, if $\omega_j^*$ is fitted by two components, then we have $|\mathcal{A}_j| = 2$ and $\bar{r}(|\mathcal{A}_j|) = 4$. Please refer to Lemma 1 for more details of the values of function $\bar{r}$.

| Setting | Loss Function | $g_{G_*}(Y|X)$ | $\exp(\beta_{0j}^*)$ | $\beta_{1j}^*, b_j^*$ | $a_j^*, \sigma_j^*$ |
|---|---|---|---|---|---|
| Exact-fitted | $\mathcal{D}_1$ | $\mathcal{O}(n^{-1/2})$ | $\mathcal{O}(n^{-1/2})$ | $\mathcal{O}(n^{-1/2})$ | $\mathcal{O}(n^{-1/2})$ |
| Over-fitted | $\mathcal{D}_2$ | $\mathcal{O}(n^{-1/2})$ | $\mathcal{O}(n^{-1/2})$ | $\mathcal{O}(n^{-1/2\bar{r}(|\mathcal{A}_j|)})$ | $\mathcal{O}(n^{-1/\bar{r}(|\mathcal{A}_j|)})$ |

**Practical implications:** Although the slow rates of the MLE under the over-fitted settings of the softmax gating Gaussian mixture of experts may seem discouraging, a practical implication of these results is that we should not choose the number of experts $k$ to be very large compared to the true number of experts $k_*$. Furthermore, the slow rates can also be useful for post-processing procedures, such as merge-truncate-merge procedure [24], with the MLE to reduce the number of experts so as to consistently estimate $k_*$ when the number of data is sufficiently large. In particular, an important insight from the theoretical results is that we can merge the MLE parameters that are close and within the range of their rates of convergence or truncate the parameters that lead to small weights of the experts. As the sample size becomes sufficiently large, the reduced number of experts may converge to the true number of experts. We leave an investigation of such model selection with the Gaussian mixture of experts via the rates of MLE for future work.

**Organization:** The paper is organized as follows. In Section 2, we first provide background on the identifiability and rate of conditional density estimation in the softmax gating Gaussian mixture of experts. Next, we proceed to establish the convergence rate of the MLE under both the exact-fitted and over-fitted settings of these models in Section 3. Then, we conclude the paper with a few discussions in Section 4. Finally, full proofs of the results and a simulation study are provided in the Appendices.

**Notation:** Firstly, we denote $[n] := \{1, 2, \ldots, n\}$ for any positive integer $n$. Next, for any vector $u \in \mathbb{R}^d$ and $z := (z_1, z_2, \ldots, z_d) \in \mathbb{N}^d$, we denote $u^z = u_1^{z_1} u_2^{z_2} \ldots u_d^{z_d}$, $|u| := u_1 + u_2 + \ldots + u_d$ and $z! := z_1! z_2! \ldots z_d!$, while $\|u\|$ represents for its 2-norm value. Additionally, the notation $|A|$ indicates the cardinality of any set $A$. Given any two positive sequences $\{a_n\}_{n \geq 1}$ and $\{b_n\}_{n \geq 1}$, we write $a_n = \mathcal{O}(b_n)$ or $a_n \lesssim b_n$ if $a_n \leq C b_n$ for all $n \in \mathbb{N}$, where $C > 0$ is some universal constant. Lastly, for any two probability density functions $p, q$ dominated by the Lebesgue measure $\mu$, we denote

$h^2(p,q) = \frac{1}{2}\int(\sqrt{p}-\sqrt{q})^2 d\mu$ as the their squared Hellinger distance and $V(p,q) = \frac{1}{2}\int|p-q|d\mu$ as their Total Variation distance.

## 2 Background

In this section, we begin with the following result on the identifiability of the softmax gating Gaussian mixture of experts, which was previously studied in [36].

**Proposition 1** (Identifiability of the softmax gating Gaussian mixture of experts). *For any mixing measures* $G = \sum_{i=1}^{k}\exp(\beta_{0i})\delta_{(\beta_{1i},a_i,b_i,\sigma_i)}$ *and* $G' = \sum_{i=1}^{k'}\exp(\beta'_{0i})\delta_{(\beta'_{1i},a'_i,b'_i,\sigma'_i)}$, *if we have* $g_G(Y|X) = g_{G'}(Y|X)$ *for almost surely* $(X,Y)$, *then it follows that* $k = k'$ *and* $G \equiv G'_{t_1,t_2}$ *where* $G'_{t_1,t_2} := \sum_{i=1}^{k'}\exp(\beta'_{0i}+t_1)\delta_{(\beta'_{1i}+t_2,a'_i,b'_i,\sigma'_i)}$ *for some* $t_1 \in \mathbb{R}$ *and* $t_2 \in \mathbb{R}^d$.

Proof of Proposition 1 is in Appendix B.1. The identifiability of the softmax gating Gaussian mixture of experts guarantees that the MLE $\widehat{G}_n$ (2) converges to the true mixing measure $G_*$ (up to the translation of the parameters in the softmax gating).

Given the consistency of the MLE, it is natural to ask about its convergence rate to the true parameters. Our next result establishes the convergence rate of conditional density estimation $g_{\widehat{G}_n}(Y|X)$ to the true conditional density $g_{G_*}(Y|X)$, which lays an important foundation for the study of MLE's convergence rate.

**Proposition 2** (Density estimation rate). *Given the MLE in equation* (2)*, the conditional density estimation* $g_{\widehat{G}_n}(Y|X)$ *has the following convergence rate:*

$$\mathbb{P}(\mathbb{E}_X[h(g_{\widehat{G}_n}(\cdot|X), g_{G_*}(\cdot|X))] > C(\log(n)/n)^{1/2}) \lesssim \exp(-c\log n),$$

*where* $c$ *and* $C$ *are universal constants.*

Proof of Proposition 2 is in Appendix B.2. The result of Proposition 2 indicates that under either the exact-fitted or over-fitted settings of the softmax gating Gaussian mixture of experts, the rate of the conditional density function $g_{\widehat{G}_n}(Y|X)$ to the true one $g_{G_*}(Y|X)$ under Hellinger distance is of order $\mathcal{O}(n^{-1/2})$ (up to some logarithmic factors), which is parametric on the sample size.

**From density estimation to parameter estimation:** The parametric rate of the conditional density estimation in Proposition 2 suggests that as long as we can establish the Hellinger lower bound $\mathbb{E}_X[h(g_G(\cdot|X), g_{G_*}(\cdot|X))] \gtrsim \mathcal{D}(G, G_*)$ for any mixing measure $G \in \mathcal{O}_k(\Theta)$ for some metric $\mathcal{D}$ among the parameters, then we obtain directly the parametric convergence rate of the MLE under the metric $\mathcal{D}$. Therefore, the main focus of the next section is to determine such metric $\mathcal{D}$ and to establish that lower bound under either exact-fitted or over-fitted settings of the Gaussian mixture of experts.

## 3 Convergence Rate of the Maximum Likelihood Estimation

In this section, we first study the convergence rate of the MLE under the exact-fitted settings of the softmax gating Gaussian mixture of experts in Section 3.1. Then, we move to the over-fitted settings in Section 3.2. Finally, we provide a proof sketch of the theories in Section 3.3.

### 3.1 Exact-fitted Settings

For the exact-fitted settings, namely, when the chosen number of experts $k$ is equal to the true number of experts $k_*$, as we mentioned in the introduction, the proper metric between the MLE and the true mixing measure is the metric $\mathcal{D}_1$ defined in equation (4), which is given by:

$$\mathcal{D}_1(G, G_*) := \inf_{t_1,t_2} \sum_{j=1}^{k_*}\left[\sum_{i\in\mathcal{A}_j}\exp(\beta_{0i})\|(\Delta_{t_2}\beta_{1ij}, \Delta a_{ij}, \Delta b_{ij}, \Delta\sigma_{ij})\|\right.$$

$$\left. + \left|\sum_{i\in\mathcal{A}_j}\exp(\beta_{0i}) - \exp(\beta_{0j}^* + t_1)\right|\right],$$

where $\Delta_{t_2}\beta_{1ij} := \beta_{1i} - \beta_{1j}^* - t_2$, $\Delta a_{ij} := a_i - a_j^*$, $\Delta b_{ij} := b_i - b_j^*$, $\Delta\sigma_{ij} := \sigma_i - \sigma_j^*$. Here, $\mathcal{A}_j$ is a Voronoi cell of $G$ generated by $(\beta_{1j}^*, a_j^*, b_j^*, \sigma_j^*)$ for all $1 \leq j \leq k_*$. Furthermore, the infimum is taken with respect to $(t_1, t_2) \in \mathbb{R} \times \mathbb{R}^d$ such that $\beta_{0j}^* + t_1$ and $\beta_{1j}^* + t_2$ still lie inside the domain of the parameter space $\Theta$.

It is clear that $\mathcal{D}_1(G, G_*) = 0$ if and only if $G \equiv G_*$ (up to translation). When $\mathcal{D}_1(G, G_*)$ is sufficiently small, there exist $t_1, t_2$ such that all of $\Delta_{t_2}\beta_{1ij}$, $\Delta a_{ij}$, $\Delta b_{ij}$, $\Delta\sigma_{ij}$, and $\sum_{i\in\mathcal{A}_j} \exp(\beta_{0i}) - \exp(\beta_{0j}^* + t_1)$ are sufficiently small as well. Therefore, the loss function $\mathcal{D}_1$ provides a useful metric to measure the difference between the MLE and the true mixing measure. For any fixed $t_1, t_2$, the computation of the summations in $\mathcal{D}_1$ only has the complexity of the order $\mathcal{O}(k_*^2)$. To solve the optimization with respect to $t_1, t_2$ in the metric $\mathcal{D}_1$, we can utilize the projected subgradient method with fixed step size [4], which has the complexity of the order $\mathcal{O}(\varepsilon^{-2})$ as the functions of $t_1$ and $t_2$ are convex where $\varepsilon$ is a desired tolerance. Therefore, the total computational complexity of approximating the value of the Voronoi loss function $\mathcal{D}_1$ is at the order of $\mathcal{O}(k_*^2/\varepsilon^2)$.

The following result establishes the lower bound of the Hellinger distance between the conditional densities in terms of the loss function $\mathcal{D}_1$ between corresponding mixing measures, which in turn leads to the convergence rate of the MLE.

**Theorem 1.** *Given the exact-fitted settings of the softmax gating Gaussian mixture of experts* (1)*, i.e., $k = k_*$, we find that*

$$\mathbb{E}_X[h(g_G(\cdot|X), g_{G_*}(\cdot|X))] \geq C_1 \cdot \mathcal{D}_1(G, G_*), \tag{7}$$

*for any $G \in \mathcal{E}_{k_*}(\Theta) := \mathcal{O}_{k_*}(\Theta) \setminus \mathcal{O}_{k_*-1}(\Theta)$ where $C_1$ is some universal constant depending only on $G_*$ and $\Theta$. As a consequence, there exist universal constants $C_1'$ and $c_1$ such that the convergence rate of the MLE $\widehat{G}_n$ under the exact-fitted settings satisfies:*

$$\mathbb{P}(\mathcal{D}_1(\widehat{G}_n, G_*) > C_1'(\log(n)/n)^{1/2}) \lesssim \exp(-c_1 \log n). \tag{8}$$

Proof of Theorem 1 is in Appendix A.1. The parametric convergence rate of the MLE to $G_*$ under the metric $\mathcal{D}_1$ suggests that the rates of estimating the true parameters $\exp(\beta_{0j}^*), \beta_{1j}^*$ (up to translation), $a_j^*, b_j^*, \sigma_j^*$ for $j \in [k_*]$ are of order $\mathcal{O}(n^{-1/2})$, which are optimal up to logarithmic factors.

## 3.2   Over-fitted Settings

We now consider the over-fitted settings of the softmax gating Gaussian mixture of experts. Different from the exact-fitted settings, the softmax weights associated with the MLE collapse to the softmax weights of the mixture of true experts as long as the MLE approaches the true mixing measure $G_*$. More concretely, we can relabel the supports of the MLE $\widehat{G}_n$ with $\hat{k}_n$ components ($\hat{k}_n \leq k$) based on the Voronoi cells $\mathcal{A}_j^n := \mathcal{A}_j(\widehat{G}_n)$ such that we can rewrite it as $\widehat{G}_n = \sum_{j=1}^{k_*} \sum_{i\in\mathcal{A}_j^n} \exp(\widehat{\beta}_{0i}^n)\delta_{(\widehat{\beta}_{1i}^n, \widehat{a}_i^n, \widehat{b}_i^n, \widehat{\sigma}_i^n)}$ where $\sum_{j=1}^{k_*} |\mathcal{A}_j^n| = \hat{k}_n$, $(\widehat{a}_i^n, \widehat{b}_i^n, \widehat{\sigma}_i^n) \to (a_j^*, b_j^*, \sigma_j^*)$,

$$\sum_{i\in\mathcal{A}_j^n} \frac{\exp((\widehat{\beta}_{1i}^n)^\top X + \widehat{\beta}_{0i}^n)}{\sum_{j'=1}^{k_*} \sum_{i'\in\mathcal{A}_{j'}^n} \exp((\widehat{\beta}_{1i'}^n)^\top X + \widehat{\beta}_{0i'}^n)} \to \frac{\exp((\beta_{1j}^*)^\top X + \beta_{0j}^*)}{\sum_{j'=1}^{k_*} \exp((\beta_{1j'}^*)^\top X + \beta_{0j'}^*)}$$

as $n$ approaches infinity for all $1 \leq i \leq \mathcal{A}_j^n$ and $j \in [k_*]$.

The collapse of the softmax weights along with the PDEs (3) between the softmax gating and the expert functions in the Gaussian density create a complex interaction among the estimated parameters. To disentangle such interaction, we rely on the solvability of a novel system of polynomial equations defined in equation (9). In particular, for any $m \geq 2$, we define $\bar{r}(m)$ as the smallest natural number $r$ such that the following system of polynomial equations:

$$\sum_{j=1}^m \sum_{(\alpha_1,\alpha_2,\alpha_3,\alpha_4)\in\mathcal{I}_{\ell_1,\ell_2}} \frac{p_{5j}^2 \, p_{1j}^{\alpha_1} \, p_{2j}^{\alpha_2} \, p_{3j}^{\alpha_3} \, p_{4j}^{\alpha_4}}{\alpha_1! \, \alpha_2! \, \alpha_3! \, \alpha_4!} = 0, \tag{9}$$

for any $(\ell_1, \ell_2) \in \mathbb{N}^d \times \mathbb{N}$ such that $0 \leq |\ell_1| \leq r$, $0 \leq \ell_2 \leq r - |\ell_1|$ and $|\ell_1| + \ell_2 \geq 1$, does not have any non-trivial solution for the unknown variables $\{p_{1j}, p_{2j}, p_{3j}, p_{4j}, p_{5j}\}_{j=1}^m$, namely, all of $p_{5j}$ are non-zero and at least one among $p_{3j}$ is different from zero. The ranges of $\alpha_1, \alpha_2, \alpha_3, \alpha_4$ in the above

sum satisfy $\mathcal{I}_{\ell_1, \ell_2} = \{\alpha = (\alpha_1, \alpha_2, \alpha_3, \alpha_4) \in \mathbb{N}^d \times \mathbb{N}^d \times \mathbb{N} \times \mathbb{N} : \alpha_1 + \alpha_2 = \ell_1, |\alpha_2| + \alpha_3 + 2\alpha_4 = \ell_2\}$. When $d = 1$ and $r = 2$, that system of equations becomes

$$\sum_{j=1}^{m} p_{5j}^2 p_{1j} = 0, \ \sum_{j=1}^{m} p_{5j}^2 p_{1j}^2 = 0, \ \sum_{j=1}^{m} p_{5j}^2 (p_{1j}p_{3j} + p_{2j}) = 0,$$

$$\sum_{j=1}^{m} p_{5j}^2 p_{3j} = 0, \ \sum_{j=1}^{m} p_{5j}^2 \left(\frac{1}{2}p_{3j}^2 + p_{4j}\right) = 0.$$

It is clear that we have non-trivial solutions $p_{5j} = 1$, $p_{1j} = 0$ for all $j \in [m]$, $|p_{21}| = p_{31} = 1$, $|p_{22}| = p_{32} = -1$, $p_{41} = p_{42} = -1/2$, $p_{2j} = p_{3j} = p_{4j} = 0$ for $3 \le j \le m$.

When $d = 1$ and $r = 3$, the system of equations can be written as follows:

$$\sum_{j=1}^{m} p_{5j}^2 p_{1j} = 0, \quad \sum_{j=1}^{m} p_{5j}^2 p_{3j} = 0, \quad \sum_{j=1}^{m} p_{5j}^2 (p_{2j} + p_{1j}p_{3j}) = 0,$$

$$\sum_{j=1}^{m} p_{5j}^2 p_{1j}^2 = 0, \quad \sum_{j=1}^{m} p_{5j}^2 \left(\frac{1}{2}p_{3j}^2 + p_{4j}\right) = 0, \quad \sum_{j=1}^{m} p_{5j}^2 \left(\frac{1}{3!}p_{3j}^3 + p_{3j}p_{4j}\right) = 0,$$

$$\sum_{j=1}^{m} p_{5j}^2 p_{1j}^3 = 0, \quad \sum_{j=1}^{m} p_{5j}^2 \left(\frac{1}{2}p_{1j}^2 p_{3j} + p_{1j}p_{2j}\right) = 0,$$

$$\sum_{j=1}^{m} p_{5j}^2 \left(\frac{1}{2}p_{1j} \cdot p_{3j}^2 + p_{1j}p_{4j} + p_{2j}p_{3j}\right) = 0.$$

It can be seen that the following is a non-trivial solution of the above system: $p_{5j} = 1$, $p_{1j} = p_{2j} = 0$ for all $j \in [m]$, $p_{31} = \frac{\sqrt{3}}{3}$, $p_{32} = -\frac{\sqrt{3}}{3}$, $p_{41} = p_{42} = -\frac{1}{6}$, $p_{3j} = p_{4j} = 0$ for $3 \le j \le m$. Therefore, we obtain that $\bar{r}(m) \ge 4$ when $m \ge 2$ and $d = 1$.

In general, when $d = 1$, the system of equations has $(r^2 + 3r)/2$ equations. Intuitively, when $m$ is sufficiently larger than $(r^2 + 3r)/2$, the system may not have a non-trivial solution. For general dimension $d$ and parameter $m \ge 2$, finding the exact value of $\bar{r}(m)$ is a non-trivial central problem in algebraic geometry [55]. When $m$ is small, the following lemma provides specific values for $\bar{r}(m)$.

**Lemma 1.** *For any $d \ge 1$, when $m = 2$, $\bar{r}(m) = 4$. When $m = 3$, $\bar{r}(m) = 6$.*

Proof of Lemma 1 is in Appendix B.3. As $m$ increases, so does the value of $\bar{r}(m)$. We conjecture that $\bar{r}(m) = 2m$ and leave the proof of that conjecture to future work.

By constructing the Voronoi loss function:

$$\mathcal{D}_2(G, G_*) := \inf_{t_1, t_2} \sum_{j:|\mathcal{A}_j|>1} \sum_{i \in \mathcal{A}_j} \exp(\beta_{0i}) \left( \|(\Delta_{t_2}\beta_{1ij}, \Delta b_{ij})\|^{\bar{r}(|\mathcal{A}_j|)} + \|(\Delta a_{ij}, \Delta \sigma_{ij})\|^{\bar{r}(|\mathcal{A}_j|)/2} \right)$$

$$+ \sum_{j:|\mathcal{A}_j|=1} \sum_{i \in \mathcal{A}_j} \exp(\beta_{0i}) \|(\Delta_{t_2}\beta_{1ij}, \Delta a_{ij}, \Delta b_{ij}, \Delta \sigma_{ij})\| + \sum_{j=1}^{k_*} \left| \sum_{i \in \mathcal{A}_j} \exp(\beta_{0i}) - \exp(\beta_{0j}^* + t_1) \right|,$$

the following result demonstrates that the convergence rates of the MLE under the over-fitted settings of the softmax gating Gaussian mixture of experts are determined by $\bar{r}(\cdot)$.

**Theorem 2.** *Under the over-fitted settings of the softmax gating Gaussian mixture of experts* (1), *namely, when $k > k_*$, we obtain that*

$$\mathbb{E}_X[h(g_G(\cdot|X), g_{G_*}(\cdot|X))] \ge C_2 \cdot \mathcal{D}_2(G, G_*), \tag{10}$$

*for any $G \in \mathcal{O}_k(\Theta)$ where $C_2$ is some universal constant depending only on $G_*$ and $\Theta$. Therefore, that lower bound leads to the following convergence rate of the MLE:*

$$\mathbb{P}(\mathcal{D}_2(\widehat{G}_n, G_*) > C_2'(\log(n)/n)^{1/2}) \lesssim \exp(-c_2 \log n), \tag{11}$$

*where $C_2'$ and $c_2$ are some universal constants.*

Proof of Theorem 2 is in Appendix A.2. A few comments with the result of Theorem 2 are in order.

**(i) Rates of individual parameters:** The convergence rate $\mathcal{O}(n^{-1/2})$ (up to some logarithmic term) of the MLE under the loss function $\mathcal{D}_2$ implies that for the true parameters $\exp(\beta_{0j}^*), \beta_{1j}^*, a_j^*, b_j^*, \sigma_j^*$ whose Voronoi cells have only one component of the MLE, the rates for estimating them are $\mathcal{O}(n^{-1/2})$ up to some logarithmic factor. On the other hand, for true parameters with greater than one component in their Voronoi cells, the rates for estimating $\beta_{1j}^*, b_j^*$ are $\mathcal{O}(n^{-1/2\bar{r}(|\mathcal{A}_j^n|)})$ while those for $a_j^*, \sigma_j^*$ are $\mathcal{O}(n^{-1/\bar{r}(|\mathcal{A}_j^n|)})$ (up to logarithmic factors). As the maximum value of $|\mathcal{A}_j^n|$ is $\hat{k}_n - k_* + 1$, it indicates that these rates (up to logarithmic factors) can be as worse as $\mathcal{O}(n^{-1/\bar{r}(\hat{k}_n - k_* + 1)})$ for estimating $a_j^*, \sigma_j^*$ and $\mathcal{O}(n^{-1/2\bar{r}(\hat{k}_n - k_* + 1)})$ for estimating $\beta_{1j}^*, b_j^*$.

**(ii) Computation of Voronoi loss function $\mathcal{D}_2$:** Similar to the Voronoi loss function $\mathcal{D}_1$ in the exact-fitted setting, the loss function $\mathcal{D}_2$ is also computationally efficient. In particular, for any fixed $t_1, t_2$, the computation of the summations in the formulation of $\mathcal{D}_2$ is at the order $\mathcal{O}(k \times k_*)$, which is linear on $k$ when $k_*$ is fixed. Furthermore, we can solve the convex optimization problem with respect to $t_1, t_2$ with computational complexity at the order of $\mathcal{O}(\varepsilon^{-2})$ via the projected gradient descent method with fixed step size where $\varepsilon$ is the error. Therefore, the total computational complexity of approximating the Voronoi loss function $\mathcal{D}_2$ is at the order of $\mathcal{O}(k \times k_*/\varepsilon^2)$.

**(iii) Comparison with covariate-free gating network:** We would like to remark that the results being established for parameter estimation under the softmax gating network settings of over-fitted Gaussian mixture of experts are in stark difference from those under the covariate-free gating network settings of these models [30], namely, when the gating function is independent of the covariates $X$. In particular, Theorem 2 in [30] shows that when the gating networks are independent of the covariates, the convergence rates of estimating $a_j^*$ are at the order of $\mathcal{O}(n^{-1/4})$ (up to some logarithmic factor), which are independent of the number of over-fitted components. It is different from the rates of $a_j^*$ whose Voronoi cells have more than one component in the softmax gating settings, which depends on the number of components that we over-fit the Gaussian mixture of experts (see discussion (i) after Theorem 2). Furthermore, the rates of estimating $b_j^*, \sigma_j^*$ when the gating networks are independent of covariates are determined by a system of polynomial equations that is much simpler than the system of equations (9) when the gating networks are softmax function. These differences are mainly due to the intrinsic interaction characterized by partial differential equations with respect to the parameters between the softmax gating networks and the expert functions in Gaussian distribution.

### 3.3 Proof Sketch

In this section, we provide a proof sketch for Theorems 1 and 2. To simplify the ensuing discussion, the loss function $\mathcal{D}$ in the proof sketch is implicitly understood as either $\mathcal{D}_1$ or $\mathcal{D}_2$ depending on the settings of the softmax gating Gaussian mixture of experts. Since the Hellinger distance $h$ is lower bounded by the Total Variation distance $V$, to obtain the bounds in equations (7) and (10), it is sufficient to show that $\mathbb{E}_X[V(g_G(\cdot|X), g_{G_*}(\cdot|X))] \gtrsim \mathcal{D}(G, G_*)$. To establish this bound, we respectively prove its local and global versions by contradiction as follows:

**Local version:** In this part, we aim to show the following local inequality:

$$\lim_{\varepsilon \to 0} \inf_{G \in \mathcal{O}_k(\Theta): \mathcal{D}(G, G_*) \leq \varepsilon} \mathbb{E}_X[V(g_G(\cdot|X), g_{G_*}(\cdot|X))]/\mathcal{D}(G, G_*) > 0. \tag{12}$$

Assume that this claim does not hold true, that is, there exists a sequence $G_n = \sum_{i=1}^{k_n} \exp(\beta_{0i}^n) \delta_{(\beta_{1i}^n, a_i^n, b_i^n, \sigma_i^n)} \in \mathcal{O}_k(\Theta)$ such that both $\mathbb{E}_X[V(g_{G_n}(\cdot|X), g_{G_*}(\cdot|X))]/\mathcal{D}(G_n, G_*)$ and $\mathcal{D}(G_n, G_*)$ approach zero as $n$ tends to infinity. This implies that for any $j \in [k_*]$, we have $\sum_{i \in \mathcal{A}_j} \exp(\beta_{0i}^n) \to \exp(\beta_{0j}^*)$ and $(\beta_{1i}^n, a_i^n, b_i^n, \sigma_i^n) \to (\beta_{1j}^*, a_j^*, b_j^*, \sigma_j^*)$ and for all $i \in \mathcal{A}_j$. For the sake of presentation, we simplify the loss function $\mathcal{D}$ by assuming that it is minimized when $t_1 = 0$ and $t_2 = \mathbf{0}_d$. Now, we decompose the quantity

$$Q_n = \left[ \sum_{j'=1}^{k_*} \exp((\beta_{1j'}^*)^\top X + \beta_{0j'}^*) \right] \cdot [g_{G_n}(Y|X) - g_{G_*}(Y|X)]$$

as follows:

$$Q_n = \sum_{j=1}^{k_*} \sum_{i \in \mathcal{A}_j} \exp(\beta_{0i}^n) \Big[ u(Y|X; \beta_{1i}^n, a_i^n, b_i^n, \sigma_i^n) - u(Y|X; \beta_{1j}^*, a_j^*, b_j^*, \sigma_j^*) - v(Y|X; \beta_{1i}^n)$$

$$+ v(Y|X; \beta_{1j}^*) \Big] + \sum_{j=1}^{k_*} \Big( \sum_{i \in \mathcal{A}_j} \exp(\beta_{0i}^n) - \exp(\beta_{0j}^*) \Big) \Big[ u(Y|X; \beta_{0j}^n, a_j^*, b_j^*, \sigma_j^*) - v(Y|X; \beta_{1j}^*) \Big],$$

where we define $u(Y|X; \beta_1, a, b, \sigma) := \exp(\beta_1^\top X) f(Y|a^\top X + b, \sigma)$ and $v(Y|X; \beta_1) := \exp(\beta_1^\top X) g_{G_n}(Y|X)$. Next, for each $j \in [k_*]$ and $i \in \mathcal{A}_j$, we denote $h_1(X, a_j^*, b_j^*) := (a_j^*)^\top X + b_j^*$ and then apply the Taylor expansions to the functions $u(Y|X; \beta_{1i}^n, a_i^n, b_i^n, \sigma_i^n)$ and $v(Y|X; \beta_{1i}^n)$ up to orders $r_{1j}$ and $r_{2j}$ (which we will choose later), respectively, as follows:

$$u(Y|X; \beta_{1i}^n, a_i^n, b_i^n, \sigma_i^n) - u(Y|X; \beta_{1j}^*, a_j^*, b_j^*, \sigma_j^*)$$

$$= \sum_{|\ell_1| + \ell_2 = 1}^{2r_{1j}} T_{\ell_1, \ell_2}^n(j) X^{\ell_1} \exp((\beta_{1j}^*)^\top X) \frac{\partial^{\ell_2} f}{\partial h_1^{\ell_2}} (Y|(a_j^*)^\top X + b_j^*, \sigma_j^*) + R_{1ij}(X, Y),$$

$$v(Y|X; \beta_{1i}^n) - v(Y|X; \beta_{1j}^*) = \sum_{|\gamma| = 1}^{r_{2j}} S_\gamma^n(j) X^\gamma \exp((\beta_{1j}^*)^\top X) g_{G_n}(Y|X) + R_{2ij}(X, Y),$$

where $R_{1ij}(X, Y)$ and $R_{2ij}(X, Y)$ are Taylor remainders such that $R_{\rho ij}(X, Y)/\mathcal{D}(G_n, G_*)$ vanishes as $n \to \infty$ for $\rho \in \{1, 2\}$. As a result, the limit of $Q_n/\mathcal{D}(G_n, G_*)$ when $n$ goes to infinity can be seen as a linear combination of elements of the following set:

$$\mathcal{W} := \left\{ X^{\ell_1} \exp((\beta_{1j}^*)^\top X) \frac{\partial^{\ell_2} f}{\partial h_1^{\ell_2}} (Y|(a_j^*)^\top X + b_j^*, \sigma_j^*) : j \in [k_*], \ 0 \le 2|\ell_1| + \ell_2 \le 2r_{1j} \right\}$$

$$\cup \left\{ X^\gamma \exp((\beta_{1j}^*)^\top X) g_{G_*}(Y|X) : j \in [k_*], \ 0 \le |\gamma| \le r_{2j} \right\},$$

which is shown to be linearly independent. By the Fatou's lemma, we demonstrate that $Q_n/\mathcal{D}(G_n, G_*)$ goes to zero as $n \to \infty$, implying that all the coefficients in the representation of $Q_n/\mathcal{D}(G_n, G_*)$, denoted by $T_{\ell_1, \ell_2}^n(j)/\mathcal{D}(G_n, G_*)$ and $S_\gamma^n(j)/\mathcal{D}(G_n, G_*)$, vanish when $n \to \infty$. Given that result, we aim to select the Taylor orders $r_{1j}$ and $r_{2j}$ such that at least one among the limits of $T_{\ell_1, \ell_2}^n(j)/\mathcal{D}(G_n, G_*)$ and $S_\gamma^n(j)/\mathcal{D}(G_n, G_*)$ is different from zero, which leads to a contradiction. Hence, we obtain the local version of the desired inequality.

Below are the details of choosing appropriate Taylor orders in each setting.

**Exact-fitted settings:** Under this setting, since $k_*$ is known, each of the Voronoi cells $\mathcal{A}_j$ for $j \in [k_*]$ has only one element. Thus, for any $i \in \mathcal{A}_j$, we have $\exp(\beta_{0i}^n) \to \exp(\beta_{0j}^*)$ and $(\beta_{1i}^n, a_i^n, b_i^n, \sigma_i^n) \to (\beta_{1j}^*, a_j^*, b_j^*, \sigma_j^*)$. Given that result, we will select $r_{1j} = r_{2j} = 1$ for all $j \in [k_*]$ as it suffices to show that at least one among the limits of $T_{\ell_1, \ell_2}^n(j)/\mathcal{D}(G_n, G_*)$ and $S_\gamma^n(j)/\mathcal{D}(G_n, G_*)$ is different from zero. In particular, if all of them vanished, we would take the sum of all the limits of $T_{\ell_1, \ell_2}^n(j)/\mathcal{D}(G_n, G_*)$ for $(\ell_1, \ell_2)$ such that $0 \le 2|\ell_1| + \ell_2 \le 2$, which leads to a contradiction that $1 = \mathcal{D}(G_n, G_*)/\mathcal{D}(G_n, G_*) \to 0$.

**Over-fitted settings:** As $k_*$ becomes unknown in this scenario, we need higher Taylor orders to obtain the same result as in the exact-fitted setting. We will reuse the proof by contradiction method to find out those orders. More specifically, assume that all the limits of $T_{\ell_1, \ell_2}^n(j)/\mathcal{D}(G_n, G_*)$ and $S_\gamma^n(j)/\mathcal{D}(G_n, G_*)$ equal zero. After some steps of considering typical limits as in the previous setting which requires $r_{2j} = 2$ for all $j \in [k_*]$, we encounter the following system of polynomial equations:

$$\sum_{i \in \mathcal{A}_j} \sum_{(\alpha_1, \alpha_2, \alpha_3, \alpha_4) \in \mathcal{I}_{\ell_1, \ell_2}} \frac{p_{5i}^2 \, p_{1i}^{\alpha_1} \, p_{2i}^{\alpha_2} \, p_{3i}^{\alpha_3} \, p_{4i}^{\alpha_4}}{\alpha_1! \, \alpha_2! \, \alpha_3! \, \alpha_4!} = 0,$$

for all $(\ell_1, \ell_2) \in \mathbb{N}^d \times \mathbb{N}$ such that $0 \le |\ell_1| \le r_{1j}$, $0 \le \ell_2 \le r_{1j} - |\ell_1|$ and $|\ell_1| + \ell_2 \ge 1$ for some $j \in [k_*]$. Due to the construction of this system, it must have at least one non-trivial

solution. Therefore, if we choose $r_{1j} = \bar{r}(|\mathcal{A}_j|)$ for all $j \in [k_*]$, then the above system does not admit any non-trivial solutions, which leads to a contradiction. Hence, we obtain the local inequality in equation (12), which suggests that we can find a positive constant $\varepsilon'$ such that $\inf_{G \in \mathcal{O}_k(\Theta):\mathcal{D}(G,G_*) \leq \varepsilon'} \mathbb{E}_X[V(g_G(\cdot|X), g_{G_*}(\cdot|X))]/\mathcal{D}(G, G_*) > 0$.

**Global version:** Therefore, it is sufficient to demonstrate the following global inequality:

$$\inf_{G \in \mathcal{O}_k(\Theta),\mathcal{D}(G,G_*) > \varepsilon'} \mathbb{E}_X[V(g_G(\cdot|X), g_{G_*}(\cdot|X))]/\mathcal{D}(G, G_*) > 0. \tag{13}$$

Assume that this claim is not true, then we can find a mixing measure $G' \in \mathcal{O}_k(\Theta)$ such that $g_{G'}(Y|X) = g_{G_*}(Y|X)$ for almost surely $(X, Y)$. According to Proposition 1, we get that $\mathcal{D}(G', G_*) = 0$, which contradicts the hypothesis $\mathcal{D}(G', G_*) > \varepsilon'$. These arguments hold for both exact-fitted and over-fitted settings up to some changes of notations.

Hence, the proof sketch is completed.

# 4 Discussion

In the paper, we study the convergence rates of parameter estimation under both the exact-fitted and over-fitted settings of the softmax gating Gaussian mixture of experts. We introduce novel Voronoi loss functions among parameters to resolve fundamental theoretical challenges posed by the softmax gating function, including identifiability up to the translation of parameters, the interaction between softmax weights and expert functions, and the dependence between the numerator and denominator of the conditional density function. When the true number of experts is known, we demonstrate that the rates for estimating true parameters are parametric on the sample size. On the other hand, when the true number of experts is unknown and over-specified, these estimation rates turn out to be determined by the solvability of a system of polynomial equations.

There are a few natural directions arising from the paper that we leave for furture work:

- First, our work does not consider the top-K sparse softmax gating function, which has been widely used to scale up massive deep learning architectures [68, 54, 21]. It is practically important to extend the current theories to establish the convergence rates of parameter estimation in the Gaussian mixture of experts with that gating function.

- Second, the paper only takes into account the regression settings, namely when the distribution of $Y$ is assumed to be continuous. Given that mixture of experts has also been used in classification settings [22, 31, 53, 34, 35, 60], namely when $Y$ is a discrete response variable, it is desirable to establish a comprehensive theory for parameter estimation under these settings of mixtures of experts.

- Third, the theories developed in the paper lay an important foundation for understanding parameter estimation in more complex models, including hierarchical mixture of experts [33, 51, 37, 66] and multigate mixture of experts [44, 26, 45].

- Finally, the convergence rates of the MLE in this work are established under the well-specified settings, namely when the data are drawn from the softmax gating Gaussian mixture of experts. Nevertheless, the convergence analysis of the MLE under the misspecified settings, namely when the data are not necessarily generated from that model, has remained poorly understood. Under those settings, the MLE $\widehat{G}_n$ converges to the mixing measures $\overline{G} \in \arg\min_{G \in \mathcal{O}_k(\Theta)} \mathrm{KL}(g_G(Y|X), p(Y|X))$ where $p(Y|X)$ is the true conditional density function of $Y$ given $X$, and it is not a softmax gating Gaussian mixture of experts. Additionally, the notation KL stands for the Kullback-Leibler divergence. The insights from our theories under the well-specified setting indicate that the Voronoi loss functions can be used to obtain the precise rates of individual parameters of the MLE $\widehat{G}_n$ to those of the mixing measure $\overline{G}$.

# Acknowledgements

NH acknowledges support from the NSF IFML 2019844 and the NSF AI Institute for Foundations of Machine Learning.

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

# Supplement to "Demystifying Softmax Gating Function in Gaussian Mixture of Experts"

In this supplementary material, we present the proofs of Theorems 1 and 2 in Appendix A, and then provide proofs for the remaining results in Appendix B. Finally, we carry out a simulation study to illustrate the various convergence rates that were derived in Theorems 1 and 2 in Appendix C.

## A Proofs of Main Results

In this appendix, we provide proof for Theorem 1 in Appendix A.1, while leave that for Theorem 2 in Appendix A.2. Prior to discussing in more detail, let us recall some notations for high-dimensional settings that we will use in our proofs. Firstly, for any vector $u := (u_1, u_2, \ldots, u_d) \in \mathbb{R}^d$ and $z := (z_1, z_2, \ldots, z_d) \in \mathbb{N}^d$, we denote $u^z = u_1^{z_1} u_2^{z_2} \ldots u_d^{z_d}$, $|u| := u_1 + u_2 + \ldots + u_d$ and $z! := z_1! z_2! \ldots z_d!$. Additionally, $\mathbf{0}_d$ denotes the vector zero in $\mathbb{R}^d$, whereas the notation $\mathbf{1}$ stands for the indicator function. Finally, we denote $h_1(X, a, b) = a^\top X + b$ and $h_2(X, \sigma) = \sigma$ as mean and variance expert functions in this work for any $(a, b, \sigma) \in \mathbb{R}^d \times \mathbb{R} \times \mathbb{R}_+$ and $X \in \mathcal{X}$.

### A.1 Proof of Theorem 1

**General Picture**: In this proof, we focus mainly on establishing the following bound:

$$\mathbb{E}_X[V(g_G(\cdot|X), g_{G_*}(\cdot|X))] \geq C_1 \cdot \mathcal{D}_1(G, G_*). \tag{14}$$

Then, the above bound together with the result of Proposition 2 lead to the conclusion of Theorem 1.

**Local version**: Firstly, we prove the local version of the inequality (14):

$$\lim_{\varepsilon \to 0} \inf_{G \in \mathcal{E}_{k_*}(\Theta): \mathcal{D}_1(G, G_*) \leq \varepsilon} \mathbb{E}_X[V(g_G(\cdot|X), g_{G_*}(\cdot|X))] / \mathcal{D}_1(G, G_*) > 0. \tag{15}$$

Suppose that the inequality in equation (15) does not hold true, then we can find a sequence $G_n := \sum_{i=1}^{k_*} \exp(\beta_{0i}^n) \delta_{(\beta_{1i}^n, a_i^n, b_i^n, \sigma_i^n)} \in \mathcal{E}_{k_*}(\Theta)$ such that

$$\mathbb{E}_X[V(g_{G_n}(\cdot|X), g_{G_*}(\cdot|X))] / \mathcal{D}_1(G_n, G_*) \to 0,$$
$$\mathcal{D}_1(G_n, G_*) \to 0,$$

as $n \to \infty$. Next, we consider the Voronoi cells $\mathcal{A}_j^n := \mathcal{A}_j(G_n)$, for $j \in [k_*]$, of the mixing measure $G_n$ generated by the true components of $G_*$. Since the argument in this proof is asymptotic, we assume without loss of generality (WLOG) that those Voronoi cells are independent of $n$ for all $n \in \mathbb{N}$, i.e. $\mathcal{A}_j = \mathcal{A}_j^n$. Additionally, since $k_*$ is known under the exact-fitted settings and $\mathcal{D}_1(G_n, G_*) \to 0$, the Voronoi cell $\mathcal{A}_j$ has only one element for any $j \in [k_*]$. WLOG, we assume that $\mathcal{A}_j = \{j\}$ for all $j \in [k_*]$, which follows that $(a_j^n, b_j^n, \sigma_j^n) \to (a_j^*, b_j^*, \sigma_j^*)$ as $n \to \infty$. Furthermore, there exist $t_1 \in \mathbb{R}$ and $t_2 \in \mathbb{R}^d$ independent of $n$ such that $\exp(\beta_{0j}^n) \to \exp(\beta_{0j}^* + t_1)$ and $\beta_{1j}^n \to \beta_{1j}^* + t_2$ as $n$ approaches infinity for all $j \in [k_*]$. It indicates that we can upper bound the Voronoi loss function $\mathcal{D}_1$ as $\mathcal{D}_1(G_n, G_*) \leq \mathcal{D}_1'(G_n, G_*)$, where

$$\mathcal{D}_1'(G_n, G_*) := \sum_{j=1}^{k_*} \left[ \exp(\beta_{0j}^n) \|(\Delta_{t_2} \beta_{1j}^n, \Delta a_j^n, \Delta b_j^n, \Delta \sigma_j^n)\| + \left| \exp(\beta_{0j}^n) - \exp(\beta_{0j}^* + t_1) \right| \right],$$

in which

$$\Delta_{t_2} \beta_{1j}^n := \beta_{1j}^n - \beta_{1j}^* - t_2, \quad \Delta a_j^n := a_j^n - a_j^*,$$
$$\Delta b_j^n := b_j^n - b_j^*, \quad \Delta \sigma_j^n := \sigma_j^n - \sigma_j^*.$$

As $\mathbb{E}_X[V(g_{G_n}(\cdot|X), g_{G_*}(\cdot|X))] / \mathcal{D}_1(G_n, G_*) \to 0$ when $n \to \infty$, we also obtain that

$$\mathbb{E}_X[V(g_{G_n}(\cdot|X), g_{G_*}(\cdot|X))] / \mathcal{D}_1'(G_n, G_*) \to 0.$$

**Step 1: Density Decomposition**

Subsequently, we consider $Q_n := [\sum_{j=1}^{k_*} \exp((\beta_{1j}^* + t_2)^\top X + \beta_{0j}^* + t_1)] \cdot [g_{G_n}(Y|X) - g_{G_*}(Y|X)]$, which can decomposed as

$$
Q_n = \sum_{j=1}^{k_*} \exp(\beta_{0j}^n) \Big[ u(Y|X; \beta_{1j}^n, a_j^n, b_j^n, \sigma_j^n) - u(Y|X; \beta_{1j}^* + t_2, a_j^*, b_j^*, \sigma_j^*) \Big]
$$

$$
- \sum_{j=1}^{k_*} \exp(\beta_{0j}^n) \Big[ v(Y|X; \beta_{1j}^n) - v(Y|X; \beta_{1j}^* + t_2) \Big]
$$

$$
+ \sum_{j=1}^{k_*} \Big( \exp(\beta_{0j}^n) - \exp(\beta_{0j}^* + t_1) \Big) \Big[ u(Y|X; \beta_{1j}^* + t_2, a_j^*, b_j^*, \sigma_j^*) - v(Y|X; \beta_{1j}^* + t_2) \Big],
$$

$$
:= A_n + B_n + E_n, \tag{16}
$$

where we denote $u(Y|X; \beta_1, a, b, \sigma) := \exp(\beta_1^\top X) f(Y|a^\top X + b, \sigma)$ and $v(Y|X; \beta_1) := \exp(\beta_1^\top X) g_{G_n}(Y|X)$. Next, by means of the first-order Taylor expansion, we rewrite $A_n$ as

$$
A_n = \sum_{j=1}^{k_*} \sum_{|\alpha|=1} \frac{\exp(\beta_{0j}^n)}{2^{\alpha_4} \alpha!} (\Delta_{t_2} \beta_{1j}^n)^{\alpha_1} (\Delta a_{ij}^n)^{\alpha_2} (\Delta b_j^n)^{\alpha_3} (\Delta \sigma_j^n)^{\alpha_4}
$$

$$
\times X^{\alpha_1 + \alpha_2} \exp((\beta_{1j}^* + t_2)^\top X) \cdot \frac{\partial^{|\alpha_2| + \alpha_3 + 2\alpha_4} f}{\partial h_1^{|\alpha_2| + \alpha_3 + 2\alpha_4}} (Y|(a_j^*)^\top X + b_j^*, \sigma_j^*) + R_1(X, Y)
$$

$$
= \sum_{j=1}^{k_*} \sum_{2|\ell_1| + \ell_2 = 1} \sum_{\alpha \in \mathcal{I}_{\ell_1, \ell_2}} \frac{\exp(\beta_{0j}^n)}{2^{\alpha_4} \alpha!} (\Delta_{t_2} \beta_{1j}^n)^{\alpha_1} (\Delta a_j^n)^{\alpha_2} (\Delta b_j^n)^{\alpha_3} (\Delta \sigma_j^n)^{\alpha_4}
$$

$$
\times X^{\ell_1} \exp((\beta_{1j}^* + t_2)^\top X) \cdot \frac{\partial^{\ell_2} f}{\partial h_1^{\ell_2}} (Y|(a_j^*)^\top X + b_j^*, \sigma_j^*) + R_1(X, Y), \tag{17}
$$

where $R_1(X, Y)$ is a Taylor remainder such that $R_1(X, Y)/\mathcal{D}_1'(G_n, G_*) \to 0$ as $n \to \infty$. Here, the first equality is due to the following partial differential equation for the univariate Gaussian density:

$$
\frac{\partial^{\alpha_4} f}{\partial h_2^{\alpha_4}} (Y|(a_j^*)^\top X + b_j^*, \sigma_j^*) = \frac{1}{2^{\alpha_4}} \cdot \frac{\partial^{2\alpha_4} f}{\partial h_1^{2\alpha_4}} (Y|(a_j^*)^\top X + b_j^*, \sigma_j^*),
$$

while the second equality is obtained by defining $\ell_1 = \alpha_1 + \alpha_2$, $\ell_2 = |\alpha_2| + \alpha_3 + 2\alpha_4$ and

$$
\mathcal{I}_{\ell_1, \ell_2} := \Big\{ \alpha = (\alpha_i)_{i=1}^4 \in \mathbb{N}^d \times \mathbb{N}^d \times \mathbb{N} \times \mathbb{N} : \alpha_1 + \alpha_2 = \ell_1, \; \alpha_3 + 2\alpha_4 = \ell_2 - |\alpha_2| \Big\}, \tag{18}
$$

for all $(\ell_1, \ell_2) \in \mathbb{N}^d \times \mathbb{N}$ such that $1 \leq 2|\ell_1| + \ell_2 \leq 2$. Analogously, $B_n$ can be rewritten as

$$
B_n = -\sum_{j=1}^{k_*} \sum_{|\gamma|=1} \frac{\exp(\beta_{0j}^n)}{\gamma!} (\Delta_{t_2} \beta_{1j}^n)^\gamma X^\gamma \exp((\beta_{1j}^* + t_2)^\top X) g_{G_n}(Y|X) + R_2(X, Y), \tag{19}
$$

where $R_2(X, Y)$ is a Taylor remainder such that $R_2(X, Y)/\mathcal{D}_1'(G_n, G_*) \to 0$ as $n \to \infty$. From the formulations of $A_n$, $B_n$ and $E_n$, we can represent $Q_n$ as the following linear combination

$$
Q_n = \sum_{j=1}^{k_*} \sum_{2|\ell_1| + \ell_2 = 0}^{2} T_{\ell_1, \ell_2}^n(j) \cdot X^{\ell_1} \exp((\beta_{1j}^* + t_2)^\top X) \frac{\partial^{\ell_2} f}{\partial h_1^{\ell_2}} (Y|(a_j^*)^\top X + b_j^*, \sigma_j^*)
$$

$$
+ \sum_{j=1}^{k_*} \sum_{|\gamma|=0}^{1} S_\gamma^n(j) \cdot X^\gamma \exp((\beta_{1j}^* + t_2)^\top X) g_{G_n}(Y|X) + R_1(X, Y) + R_2(X, Y),
$$

with coefficients being denoted by $T_{\ell_1, \ell_2}^n(j)$ and $S_\gamma^n(j)$ for all $j \in [k_*]$, $0 \leq 2|\ell_1| + \ell_2 \leq 2$ and $0 \leq |\gamma| \leq 1$ where

$$
T_{\ell_1, \ell_2}^n(j) = \begin{cases} \sum_{\alpha \in \mathcal{I}_{\ell_1, \ell_2}} \frac{\exp(\beta_{0j}^n)}{2^{\alpha_4} \alpha!} (\Delta_{t_2} \beta_{1j}^n)^{\alpha_1} (\Delta a_j^n)^{\alpha_2} (\Delta b_j^n)^{\alpha_3} (\Delta \sigma_j^n)^{\alpha_4}, & (\ell_1, \ell_2) \neq (\mathbf{0}_d, 0), \\[4mm] \exp(\beta_{0j}^n) - \exp(\beta_{0j}^* + t_1), & (\ell_1, \ell_2) = (\mathbf{0}_d, 0); \end{cases} \tag{20}
$$

and

$$S_\gamma^n(j) = \begin{cases} -\dfrac{\exp(\beta_{0j}^n)}{\gamma!}(\Delta_{t_2}\beta_{1j}^n)^\gamma, & |\gamma| \neq 0, \\[3mm] -\exp(\beta_{0j}^n) + \exp(\beta_{0j}^* + t_1), & |\gamma| = 0. \end{cases} \tag{21}$$

**Step 2: Non-vanishing coefficients**

Now, we will demonstrate by contradiction that at least one among terms of the forms $T_{\ell_1,\ell_2}^n(j)/\mathcal{D}_1'(G_n, G_*)$ and $S_\gamma^n(j)/\mathcal{D}_1'(G_n, G_*)$ does not approach zero. Indeed, assume that all of them vanish when $n \to \infty$, then we get

$$\frac{1}{\mathcal{D}_1'(G_n, G_*)} \cdot \sum_{j=1}^{k_*} \left| \exp(\beta_{0j}^n) - \exp(\beta_{0j}^* + t_1) \right| = \sum_{j=1}^{k_*} \frac{|T_{\mathbf{0}_d,0}^n(j)|}{\mathcal{D}_1'(G_n, G_*)} \to 0. \tag{22}$$

Similarly, by considering the limits of $T_{\ell_1,\ell_2}^n(j)/\mathcal{D}_1'(G_n, G_*)$ for all $j \in [k_*]$ and $1 \leq 2|\ell_1| + \ell_2 \leq 2$, we obtain that

$$\frac{1}{\mathcal{D}_1'(G_n, G_*)} \cdot \sum_{j=1}^{k_*} \exp(\beta_{0j}^n) \|(\Delta_{t_2}\beta_{1j}^n, \Delta a_j^n, \Delta b_j^n, \Delta\sigma_j^n)\| \to 0. \tag{23}$$

Combine the results in equations (22) and 23, we have $1 = \mathcal{D}_1'(G_n, G_*)/\mathcal{D}_1'(G_n, G_*) \to 0$, which is a contradiction. As a result, not all the limits of $T_{\ell_1,\ell_2}^n(j)/\mathcal{D}_1'(G_n, G_*)$ and $S_\gamma^n(j)/\mathcal{D}_1'(G_n, G_*)$ equal to zero.

**Step 3: Fatou's lemma involvement**

Thus, let $m_n$ be the maximum of the absolute values of those terms, we have that $1/m_n \nrightarrow \infty$. Then, the Fatou's lemma says that

$$\lim_{n\to\infty} \frac{\mathbb{E}_X[V(g_{G_n}(\cdot|X), g_{G_*}(\cdot|X))]}{m_n \cdot \mathcal{D}_1'(G_n, G_*)} \geq \int \liminf_{n\to\infty} \frac{|g_{G_n}(Y|X) - g_{G_*}(Y|X)|}{2m_n \cdot \mathcal{D}_1'(G_n, G_*)} \, \mathrm{d}(X, Y). \tag{24}$$

By assumption, the left-hand side of the above equation equals to zero, therefore, the integrand in the right-hand side also equals to zero for almost surely $(X, Y)$, which leads to the following limit: $Q_n/[m_n\mathcal{D}_1'(G_n, G_*)] \to 0$ as $n \to \infty$ for almost surely $(X, Y)$. More specifically, we have

$$\sum_{j=1}^{k_*} \sum_{2|\ell_1|+\ell_2=0}^{2} \eta_{\ell_1,\ell_2}(j) \cdot X^{\ell_1} \exp((\beta_{1j}^* + t_2)^\top X) \frac{\partial^{\ell_2} f}{\partial h_1^{\ell_2}}(Y|(a_j^*)^\top X + b_j^*, \sigma_j^*)$$

$$+ \sum_{j=1}^{k_*} \sum_{|\gamma|=0}^{1} \omega_\gamma(j) \cdot X^\gamma \exp((\beta_{1j}^* + t_2)^\top X) g_{G_*}(Y|X) = 0,$$

for almost surely $(X, Y)$, where $\eta_{\ell_1,\ell_2}(j)$ and $\omega_\gamma(j)$ are the limits of $T_{\ell_1,\ell_2}^n(j)/[m_n\mathcal{D}_1'(G_n, G_*)]$ and $S_\gamma^n(j)/[m_n\mathcal{D}_1'(G_n, G_*)]$, respectively, for all $j \in [k_*]$, $0 \leq 2|\ell_1| + \ell_2 \leq 2$ and $0 \leq |\gamma| \leq 1$. Here, at least one among $\eta_{\ell_1,\ell_2}(j)$ and $\omega_\gamma(j)$ is different from zero. On the other hand, since the set

$$\mathcal{W}_1 := \left\{ X^{\ell_1} \exp((\beta_{1j}^* + t_2)^\top X) \frac{\partial^{\ell_2} f}{\partial h_1^{\ell_2}}(Y|(a_j^*)^\top X + b_j^*, \sigma_j^*) : j \in [k_*], \ 0 \leq 2|\ell_1| + \ell_2 \leq 2 \right\}$$

$$\cup \left\{ X^\gamma \exp((\beta_{1j}^* + t_2)^\top X) g_{G_*}(Y|X) : j \in [k_*], \ 0 \leq |\gamma| \leq 1 \right\}, \tag{25}$$

is linearly independent (see Lemma 2 at the end of this proof), we obtain that $\eta_{\ell_1,\ell_2}(j) = \omega_\gamma(j) = 0$ for all $j \in [k_*]$, $0 \leq 2|\ell_1| + \ell_2 \leq 2$ and $0 \leq |\gamma| \leq 1$, which is a contradiction.

Thus, we reach the local inequality in 15, that is, there exists $\varepsilon' > 0$ that satisfies

$$\inf_{G \in \mathcal{E}_{k_*}(\Theta):\mathcal{D}_1(G, G_*)\leq\varepsilon'} \mathbb{E}_X[V(g_G(\cdot|X), g_{G_*}(\cdot|X))]/\mathcal{D}_1(G, G_*) > 0.$$

**Global version**: Thus, it suffices to prove its following global inequality:

$$\inf_{\substack{G\in\mathcal{E}_{k_*}(\Theta):\\ \mathcal{D}_1(G,G_*)>\varepsilon'}} \mathbb{E}_X[V(g_G(\cdot|X),g_{G_*}(\cdot|X))]/\mathcal{D}_1(G,G_*) > 0. \tag{26}$$

Assume by contrary that there exists a sequence $G'_n \in \mathcal{E}_{k_*}(\Theta)$ that satisfies

$$\begin{cases} \lim_{n\to\infty} \mathbb{E}_X[V(g_{G'_n}(\cdot|X),g_{G_*}(\cdot|X))]/\mathcal{D}_1(G'_n,G_*) = 0, \\ \mathcal{D}_1(G'_n,G_*) > \varepsilon'. \end{cases}$$

Therefore, we obtain that $\mathbb{E}_X[V(g_{G'_n}(\cdot|X),g_{G_*}(\cdot|X))] \to 0$ as $n \to \infty$. Since the set $\Theta$ is compact, we are able to replace the sequence $G'_n$ by its subsequence which converges to some mixing measure $G' \in \mathcal{E}_{k_*}(\Theta)$ such that $\mathcal{D}(G',G_*) > \varepsilon'$. Then, by the Fatou's lemma, we get

$$\lim_{n\to\infty} \mathbb{E}_X[V(g_{G'_n}(\cdot|X),g_{G_*}(\cdot|X))] \geq \frac{1}{2} \int \liminf_{n\to\infty} |g_{G'_n}(Y|X) - g_{G_*}(Y|X)| \, \mathrm{d}(X,Y),$$

which implies that

$$\int |g_{G'}(Y|X) - g_{G_*}(Y|X)| \mathrm{d}(X,Y) = 0$$

Thus, we obtain that $g_{G'}(Y|X) = g_{G_*}(Y|X)$ for almost surely $(X,Y)$. Now that the softmax gating Gaussian mixture of experts is identifiable up to a translation (see Proposition 1), the mixing measure $G'$ admits the form $G' = \sum_{i=1}^{k_*} \exp(\beta^*_{0\tau(i)} + t_1)\delta_{(\beta^*_{1\tau(i)}+t_2, a^*_{\tau(i)}, b^*_{\tau(i)}, \sigma^*_{\tau(i)})}$ for some $t_1 \in \mathbb{R}$ and $t_2 \in \mathbb{R}^d$, where $\tau$ is some permutation of the set $\{1,2,\ldots,k\}$. This leads to the fact that $\mathcal{D}_1(G',G_*) = 0$, which contradicts the hypothesis $\mathcal{D}_1(G',G_*) > \varepsilon' > 0$. Hence, we obtain the inequality in equation (14).

To complete the proof, we will show the previous claim regarding the independence of elements in $\mathcal{W}_1$ in the following lemma:

**Lemma 2.** *The set $\mathcal{W}_1$ defined in equation (25) is linearly independent w.r.t $X$ and $Y$.*

*Proof of Lemma 2.* Assume that the following holds for almost surely $(X,Y)$:

$$\sum_{j=1}^{k_*} \sum_{2|\ell_1|+\ell_2=0}^{2} \eta_{\ell_1,\ell_2}(j) \cdot X^{\ell_1} \exp((\beta^*_{1j}+t_2)^\top X) \frac{\partial^{\ell_2} f}{\partial h_1^{\ell_2}}(Y|(a^*_j)^\top X + b^*_j, \sigma^*_j)$$

$$+ \sum_{j=1}^{k_*} \sum_{|\gamma|=0}^{1} \omega_\gamma(j) \cdot X^\gamma \exp((\beta^*_{1j}+t_2)^\top X) g_{G_*}(Y|X) = 0,$$

where $\eta_{\ell_1,\ell_2}(j) \in \mathbb{R}$ and $\omega_\gamma(j) \in \mathbb{R}$. Then, we need to show that $\eta_{\ell_1,\ell_2}(j) = \omega_\gamma(j) = 0$, for all $j \in [k_*], 0 \leq 2|\ell_1| + \ell_2 \leq 2$ and $0 \leq |\gamma| \leq 1$. The above equation is equivalent to

$$\sum_{j=1}^{k_*} \sum_{\zeta=0}^{1} \Big[ \sum_{\ell_2=0}^{2-2\zeta} \eta_{\zeta,\ell_2}(j) \frac{\partial^{\ell_2} f}{\partial h_1^{\ell_2}}(Y|(a^*_j)^\top X + b^*_j, \sigma^*_j) + \omega_\zeta(j) g_{G_*}(Y|X) \Big] X^\zeta \exp((\beta^*_{1j}+t_2)^\top X) = 0,$$

for almost surely $(X,Y)$. Since $\beta^*_{11},\ldots,\beta^*_{1k_*}$ are $k_*$ distinct values, we get that the set $\{\exp((\beta^*_{1j}+t_2)^\top X) : j \in [k_*]\}$ is linearly independent, which implies that

$$\sum_{\zeta=0}^{1} \Big[ \sum_{\ell_2=0}^{2-2\zeta} \eta_{\zeta,\ell_2}(j) \frac{\partial^{\ell_2} f}{\partial h_1^{\ell_2}}(Y|(a^*_j)^\top X + b^*_j, \sigma^*_j) + \omega_\zeta(j) g_{G_*}(Y|X) \Big] X^\zeta = 0,$$

for all $j \in [k_*]$ for almost surely $(X,Y)$. Obviously, the above equation is a polynomial of $X \in \mathcal{X}$, where $\mathcal{X}$ is a compact subset of $\mathbb{R}^d$. Then, we achieve that

$$\sum_{\ell_2=0}^{2-2\zeta} \eta_{\zeta,\ell_2}(j) \frac{\partial^{\ell_2} f}{\partial h_1^{\ell_2}}(Y|(a^*_j)^\top X + b^*_j, \sigma^*_j) + \omega_\zeta(j) g_{G_*}(Y|X) = 0,$$

for all $j \in [k_*]$ and $\zeta \in \{0, 1\}$, for almost surely $(X, Y)$. Again, as $(a_j^*, b_j^*, \sigma_j^*)$ for $j \in [k_*]$ are $k_*$ distinct tuples, we have that $((a_j^*)^\top X + b_j^*, \sigma_j^*)$ for $j \in [k_*]$ are also $k_*$ distinct tuples for almost surely $X$. Therefore, $\left\{ \frac{\partial^{\ell_2} f}{\partial h_1^{\ell_2}}(Y|(a_j^*)^\top X + b_j^*, \sigma_j^*), \ g_{G_*}(Y|X) \right\}$ is a linearly independent set. As a result, $\eta_{\ell_1, \ell_2}(j) = \omega_\gamma(j) = 0$ for all $j \in [k_*]$, $0 \le 2|\ell_1| + \ell_2 \le 2$ and $0 \le |\gamma| \le 1$.

Hence, the proof is completed. $\qquad\square$

## A.2  Proof of Theorem 2

In this proof, we adapt the framework in Appendix A.1 to the setting of Theorem 2. However, since the arguments utilized for the global version part remain the same (up to some changes of notations) for the over-fitted settings, they will not be presented here again. Thus, we focus only on proving the following local inequality:

$$\lim_{\varepsilon \to 0} \inf_{\substack{G \in \mathcal{O}_k(\Theta): \\ \mathcal{D}_2(G, G_*) \le \varepsilon}} \mathbb{E}_X[V(g_G(\cdot|X), g_{G_*}(\cdot|X))]/\mathcal{D}_2(G, G_*) > 0. \qquad (27)$$

Assume that the above claim is not true, then there exists a sequence of mixing measures $G_n := \sum_{i=1}^{k_n} \exp(\beta_{0i}^n) \delta_{(\beta_{1i}^n, a_i^n, b_i^n, \sigma_i^n)} \in \mathcal{O}_k(\Theta)$ such that

$$\mathbb{E}_X[V(g_{G_n}(\cdot|X), g_{G_*}(\cdot|X))]/\mathcal{D}_2(G_n, G_*) \to 0,$$
$$\mathcal{D}_2(G_n, G_*) \to 0,$$

when $n$ tends to infinity. Since the proof argument is asymptotic, we also assume that $k_n = k'$ for all $n \ge 1$. Following the proof argument of Theorem 1 in Appendix A.1, we also assume that the Voronoi cells $\mathcal{A}_j = \mathcal{A}_j^n$ does not change with $n$ for all $j \in [k_*]$. Additionally, since $\mathcal{D}_2(G_n, G_*) \to 0$, we have $(a_i^n, b_i^n, \sigma_i^n) \to (a_j^*, b_j^*, \sigma_j^*)$ for any $i \in \mathcal{A}_j$ as $n$ approaches infinity. Furthermore, there exist $t_1 \in \mathbb{R}$ and $t_2 \in \mathbb{R}^d$ such that $\sum_{i \in \mathcal{A}_j} \exp(\beta_{0i}^n) \to \exp(\beta_{0j}^* + t_1)$ and $\beta_{1i}^n \to \beta_{1j}^* + t_2$ for any $i \in \mathcal{A}_j$ and $j \in [k_*]$. Then, we can upper bound the Voronoi loss function $\mathcal{D}_2$ as $\mathcal{D}_2(G_n, G_*) \le \mathcal{D}_2'(G_n, G_*)$, where

$$\mathcal{D}_2'(G_n, G_*) := \sum_{j: |\mathcal{A}_j| > 1} \sum_{i \in \mathcal{A}_j} \exp(\beta_{0i}^n) \Big( \|(\Delta_{t_2} \beta_{1ij}^n, \Delta b_{ij}^n)\|^{\bar{r}(|\mathcal{A}_j|)} + \|(\Delta a_{ij}^n, \Delta \sigma_{ij}^n)\|^{\bar{r}(|\mathcal{A}_j|)/2} \Big)$$

$$+ \sum_{j: |\mathcal{A}_j| = 1} \sum_{i \in \mathcal{A}_j} \exp(\beta_{0i}^n) \|(\Delta_{t_2} \beta_{1ij}, \Delta a_{ij}^n, \Delta b_{ij}^n, \Delta \sigma_{ij}^n)\| + \sum_{j=1}^{k_*} \Big| \sum_{i \in \mathcal{A}_j} \exp(\beta_{0i}^n) - \exp(\beta_{0j}^* + t_1) \Big|,$$

in which

$$\Delta_{t_2} \beta_{1ij}^n := \beta_{1i}^n - \beta_{1j}^* - t_2, \quad \Delta a_{ij}^n := a_i^n - a_j^*,$$
$$\Delta b_{ij}^n := b_i^n - b_j^*, \quad \Delta \sigma_{ij}^n := \sigma_i^n - \sigma_j^*.$$

Recall that $\mathbb{E}_X[V(g_{G_n}(\cdot|X), g_{G_*}(\cdot|X))]/\mathcal{D}_2(G_n, G_*) \to 0$ as $n \to \infty$, which leads to

$$\mathbb{E}_X[V(g_{G_n}(\cdot|X), g_{G_*}(\cdot|X))]/\mathcal{D}_2'(G_n, G_*) \to 0.$$

### Step 1: Density Decomposition

In this step, we decompose the quantity $Q_n = [\sum_{j=1}^{k_*} \exp((\beta_{1j}^* + t_2)^\top X + \beta_{0j}^* + t_1)] \cdot [g_{G_n}(Y|X) - g_{G_*}(Y|X)]$ with abuse of notations in Appendix A.1 as follows:

$$Q_n = \sum_{j=1}^{k_*} \sum_{i \in \mathcal{A}_j} \exp(\beta_{0i}^n) \Big[ u(Y|X; \beta_{1i}^n, a_i^n, b_i^n, \sigma_i^n) - u(Y|X; \beta_{1j}^* + t_2, a_j^*, b_j^*, \sigma_j^*) \Big]$$

$$- \sum_{j=1}^{k_*} \sum_{i \in \mathcal{A}_j} \exp(\beta_{0i}^n) \Big[ v(Y|X; \beta_{1i}^n) - v(Y|X; \beta_{1j}^* + t_2) \Big]$$

$$+ \sum_{j=1}^{k_*} \Big( \sum_{i \in \mathcal{A}_j} \exp(\beta_{0i}^n) - \exp(\beta_{0j}^* + t_1) \Big) \Big[ u(Y|X; \beta_{1j}^* + t_2, a_j^*, b_j^*, \sigma_j^*) - v(Y|X; \beta_{1j}^* + t_2) \Big],$$

$$:= A_n + B_n + E_n,$$

where we denote $u(Y|X; \beta_1, a, b, \sigma) := \exp(\beta_1^\top X)f(Y|a^\top X + b, \sigma)$ and $v(Y|X; \beta_1) := \exp(\beta_1^\top X)g_{G_n}(Y|X)$. Since each Voronoi cell $\mathcal{A}_j$ possibly has more than one element, we continue to decompose $A_n$ and $B_n$ as follows:

$$A_n = \sum_{j:|\mathcal{A}_j|=1} \sum_{i\in\mathcal{A}_j} \exp(\beta_{0i}^n)\Big[u(Y|X; \beta_{1i}^n, a_i^n, b_i^n, \sigma_i^n) - u(Y|X; \beta_{1j}^* + t_2, a_j^*, b_j^*, \sigma_j^*)\Big]$$

$$+ \sum_{j:|\mathcal{A}_j|>1} \sum_{i\in\mathcal{A}_j} \exp(\beta_{0i}^n)\Big[u(Y|X; \beta_{1i}^n, a_i^n, b_i^n, \sigma_i^n) - u(Y|X; \beta_{1j}^* + t_2, a_j^*, b_j^*, \sigma_j^*)\Big]$$

$$:= A_{n,1} + A_{n,2},$$

and

$$B_n = - \sum_{j:|\mathcal{A}_j|=1} \sum_{i\in\mathcal{A}_j} \exp(\beta_{0i}^n)\Big[v(Y|X; \beta_{1i}^n) - v(Y|X; \beta_{1j}^* + t_2)\Big]$$

$$- \sum_{j:|\mathcal{A}_j|>1} \sum_{i\in\mathcal{A}_j} \exp(\beta_{0i}^n)\Big[v(Y|X; \beta_{1i}^n) - v(Y|X; \beta_{1j}^* + t_2)\Big]$$

$$:= B_{n,1} + B_{n,2}.$$

Now, we apply the first-order Taylor expansions to two terms $A_{n,1}$ and $B_{n,1}$ as in equations (17) and (19), while for $A_{n,2}$ and $B_{n,2}$, we use the Taylor expansions of orders $\bar{r}(|\mathcal{A}_j|)$ and 2, respectively, for each $j : |\mathcal{A}_j| > 1$ as follows:

$$A_{n,2} = \sum_{j:|\mathcal{A}_j|>1} \sum_{i\in\mathcal{A}_j} \sum_{2|\ell_1|+\ell_2=1}^{2\bar{r}(|\mathcal{A}_j|)} \sum_{\alpha\in\mathcal{I}_{\ell_1,\ell_2}} \frac{\exp(\beta_{0i}^n)}{2^{\alpha_4}\alpha!}(\Delta_{t_2}\beta_{1ij}^n)^{\alpha_1}(\Delta a_{ij}^n)^{\alpha_2}(\Delta b_{ij}^n)^{\alpha_3}(\Delta\sigma_{ij}^n)^{\alpha_4}$$

$$\times X^{\ell_1}\exp((\beta_{1j}^* + t_2)^\top X) \cdot \frac{\partial^{\ell_2} f}{\partial h_1^{\ell_2}}(Y|(a_j^*)^\top X + b_j^*, \sigma_j^*) + R_3(X,Y),$$

$$B_{n,2} = - \sum_{j:|\mathcal{A}_j|>1} \sum_{i\in\mathcal{A}_j} \sum_{|\gamma|=1}^{2} \frac{\exp(\beta_{0i}^n)}{\gamma!}(\Delta_{t_2}\beta_{1ij}^n)^\gamma X^\gamma \exp((\beta_{1j}^* + t_2)^\top X)g_{G_n}(Y|X) + R_4(X,Y).$$

Here, $\mathcal{I}_{\ell_1,\ell_2} := \big\{\alpha = (\alpha_i)_{i=1}^4 \in \mathbb{N}^d \times \mathbb{N}^d \times \mathbb{N} \times \mathbb{N} : \alpha_1 + \alpha_2 = \ell_1, \alpha_3 + 2\alpha_4 = \ell_2 - |\alpha_2|\big\}$ for any $(\ell_1, \ell_2) \in \mathbb{N}^2$ such that $1 \leq 2|\ell_1| + \ell_2 \leq 2\bar{r}(|\mathcal{A}_j|)$, while $R_3(X,Y)$ and $R_4(X,Y)$ are Taylor remainders such that $R_p(X,Y)/\mathcal{D}_2'(G_n, G_*) \to 0$ when $n \to \infty$ for $p \in \{3, 4\}$.

As a result, $Q_n$ can be represented as

$$Q_n = \sum_{j=1}^{k_*} \sum_{2|\ell_1|+\ell_2=0}^{2\bar{r}(|\mathcal{A}_j|)} T_{\ell_1,\ell_2}^n(j) \cdot X^{\ell_1}\exp((\beta_{1j}^* + t_2)^\top X)\frac{\partial^{\ell_2} f}{\partial h_1^{\ell_2}}(Y|(a_j^*)^\top X + b_j^*, \sigma_j^*)$$

$$+ \sum_{j=1}^{k_*} \sum_{|\gamma|=0}^{1+\mathbf{1}_{\{|\mathcal{A}_j|>1\}}} S_\gamma^n(j) \cdot X^\gamma \exp((\beta_{1j}^* + t_2)^\top X)g_{G_n}(Y|X) + \sum_{\rho=1}^{4} R_\rho(X,Y), \qquad (28)$$

with coefficients $T_{\ell_1,\ell_2}^n(j)$ and $S_\gamma^n(j)$ being defined for any $j \in [k_*]$, $0 \leq 2|\ell_1| + \ell_2 \leq 2$ and $0 \leq |\gamma| \leq 1$ as

$$T_{\ell_1,\ell_2}^n(j) = \begin{cases} \sum_{i\in\mathcal{A}_j}\sum_{\alpha\in\mathcal{I}_{\ell_1,\ell_2}} \frac{\exp(\beta_{0i}^n)}{2^{\alpha_4}\alpha!}(\Delta_{t_2}\beta_{1ij}^n)^{\alpha_1}(\Delta a_{ij}^n)^{\alpha_2}(\Delta b_{ij}^n)^{\alpha_3}(\Delta\sigma_{ij}^n)^{\alpha_4}, & (\ell_1,\ell_2) \neq (\mathbf{0}_d, 0), \\[2em] \sum_{i\in\mathcal{A}_j}\exp(\beta_{0i}^n) - \exp(\beta_{0j}^* + t_1), & (\ell_1,\ell_2) = (\mathbf{0}_d, 0); \end{cases}$$

and

$$S_\gamma^n(j) = \begin{cases} -\sum_{i\in\mathcal{A}_j} \frac{\exp(\beta_{0i}^n)}{\gamma!}(\Delta_{t_2}\beta_{1ij}^n)^\gamma, & |\gamma| \neq 0, \\[2em] -\sum_{i\in\mathcal{A}_j}\exp(\beta_{0i}^n) + \exp(\beta_{0j}^* + t_1), & |\gamma| = 0. \end{cases}$$

**Step 2: Non-vanishing coefficients**

Next, we will show that not all the quantities $T^n_{\ell_1,\ell_2}(j)/\mathcal{D}'_2(G_n,G_*)$ and $S^n_\gamma(j)/\mathcal{D}'_2(G_n,G_*)$ go to 0 as $n \to \infty$. Assume that all of them vanish when $n$ tends to infinity. Then, by arguing similarly as in equations (22) and (23), we obtain that

$$
\frac{1}{\mathcal{D}'_2(G_n,G_*)} \cdot \left[ \sum_{j=1}^{k_*} \Big| \sum_{i\in\mathcal{A}_j} \exp(\beta^n_{0i}) - \exp(\beta^*_{0j}+t_1) \Big| \right.
$$
$$
\left. + \sum_{j:|\mathcal{A}_j|=1} \sum_{i\in\mathcal{A}_j} \exp(\beta^n_{0i}) \| (\Delta_{t_2}\beta^n_{1ij}, \Delta a^n_{ij}, \Delta b^n_{ij}, \Delta\sigma^n_{ij}) \| \right] \to 0.
$$

Putting the above limit and the formulation of $\mathcal{D}_2(G_n,G_*)$ together, we deduce that

$$
\frac{1}{\mathcal{D}'_2(G_n,G_*)} \cdot \sum_{j:|\mathcal{A}_j|>1} \sum_{i\in\mathcal{A}_j} \exp(\beta_{0i}) \Big( \|(\Delta_{t_2}\beta^n_{1ij}, \Delta b^n_{ij})\|^{\bar{r}(|\mathcal{A}_j|)} + \|(\Delta a^n_{ij}, \Delta\sigma^n_{ij})\|^{\bar{r}(|\mathcal{A}_j|)/2} \Big) \not\to 0,
$$

which indicates that there exists some index $j^* \in [k_*]$ such that $|\mathcal{A}_{j^*}| > 1$ and

$$
\frac{1}{\mathcal{D}'_2(G_n,G_*)} \cdot \sum_{i\in\mathcal{A}_{j^*}} \exp(\beta_{0i}) \Big( \|(\Delta_{t_2}\beta^n_{1ij^*}, \Delta b^n_{ij^*})\|^{\bar{r}(|\mathcal{A}_j|)} + \|(\Delta a^n_{ij^*}, \Delta\sigma^n_{ij^*})\|^{\bar{r}(|\mathcal{A}_j|)/2} \Big) \not\to 0,
$$

for all $t_2 \in \mathbb{R}^d$. Without loss of generality, we may assume that $j^* = 1$. Recall that for $(\ell_1,\ell_2) \in \mathbb{N}^d \times \mathbb{N}$ such that $1 \leq |\ell_1| + \ell_2 \leq \bar{r}(|\mathcal{A}_1|)$, we have $T^n_{\ell_1,\ell_2}(1)/\mathcal{D}'_2(G_n,G_*) \to 0$ as $n \to \infty$. Thus, by dividing this ratio and the left hand side of the above equation and let $t_2 = 0$, we obtain that

$$
\frac{\sum_{i\in\mathcal{A}_1} \sum_{\alpha\in\mathcal{I}_{\ell_1,\ell_2}} \frac{\exp(\beta^n_{0i})}{2^{\alpha_4}\alpha!} (\Delta_{t_2}\beta^n_{1i1})^{\alpha_1} (\Delta a^n_{i1})^{\alpha_2} (\Delta b^n_{i1})^{\alpha_3} (\Delta\sigma^n_{i1})^{\alpha_4}}{\sum_{i\in\mathcal{A}_1} \exp(\beta^n_{0i}) \Big( \|(\Delta_{t_2}\beta^n_{1i1}, \Delta b^n_{i1})\|^{\bar{r}(|\mathcal{A}_1|)} + \|(\Delta a^n_{i1}, \Delta\sigma^n_{i1})\|^{\bar{r}(|\mathcal{A}_1|)/2} \Big)} \to 0, \qquad (29)
$$

for all $(\ell_1,\ell_2)$ such that $1 \leq |\ell_1| + \ell_2 \leq \bar{r}(|\mathcal{A}_1|)$.

Let us define $\overline{M}_n := \max\{\|\Delta_{t_2}\beta^n_{1i1}\|, \|\Delta a^n_{i1}\|^{1/2}, |\Delta b^n_{i1}|, |\Delta\sigma^n_{i1}|^{1/2} : i \in \mathcal{A}_1\}$ and $\overline{\beta}_n := \max_{i\in\mathcal{A}_1} \exp(\beta^n_{0i})$. Note that the sequence $\exp(\beta^n_{0i})/\overline{\beta}_n$ is bounded, therefore, we can replace it by its subsequence that has a positive limit $p^2_{5i} := \lim_{n\to\infty} \exp(\beta^n_{0i})/\overline{\beta}_n$. Thus, at least one among $p^2_{5i}$, for $i \in \mathcal{A}_1$, equals 1.

In addition, we also define

$$
(\Delta_{t_2}\beta^n_{1i1})/\overline{M}_n \to p_{1i}, \quad (\Delta a^n_{i1})/\overline{M}_n \to p_{2i},
$$
$$
(\Delta b^n_{i1})/\overline{M}_n \to p_{3i}, \quad (\Delta\sigma^n_{i1})/[2\overline{M}_n] \to p_{4i}.
$$

Here, at least one of $p_{1i}$, $p_{2i}$, $p_{3i}$ and $p_{4i}$ for $i \in \mathcal{A}_1$ equals either 1 or $-1$. Next, we divide both the numerator and the denominator of the ratio in equation (29) by $\overline{\beta}_n \overline{M}_n^{\ell_1+\ell_2}$, and then achieve the following system of polynomial equations:

$$
\sum_{i\in\mathcal{A}_1} \sum_{\alpha\in\mathcal{I}_{\ell_1,\ell_2}} \frac{1}{\alpha!} \cdot p^2_{5i} p^{\alpha_1}_{1i} p^{\alpha_2}_{2i} p^{\alpha_3}_{3i} p^{\alpha_4}_{4i} = 0,
$$

for all $(\ell_1,\ell_2) \in \mathbb{N}^d \times \mathbb{N}$ such that $1 \leq |\ell_1| + \ell_2 \leq \bar{r}(|\mathcal{A}_1|)$. However, based on the definition of $\bar{r}(|\mathcal{A}_1|)$, the above system has no non-trivial solutions, which is a contradiction. Thus, not all the quantities $T^n_{\ell_1,\ell_2}(j)/\mathcal{D}'_2(G_n,G_*)$ and $S^n_\gamma(j)/\mathcal{D}'_2(G_n,G_*)$ go to 0 as $n \to \infty$.

**Step 3: Fatou's lemma involvement**

Subsequently, we denote by $m_n$ be the maximum of the absolute values of those quantities. Based on the result in Step 2, we know that $1/m_n \not\to \infty$. Then, by applying the Fatou's lemma as in equation (24), we get that $Q_n/[m_n\mathcal{D}'_2(G_n,G_*)] \to 0$ as $n \to \infty$ for almost surely $(X, Y)$. It follows

from the decomposition of $Q_n$ in equation (28) that

$$\sum_{j=1}^{k_*} \sum_{2|\ell_1|+\ell_2=0}^{2\bar{r}(|\mathcal{A}_j|)} \eta_{\ell_1,\ell_2}(j) \cdot X^{\ell_1} \exp((\beta_{1j}^* + t_2)^\top X) \frac{\partial^{\ell_2} f}{\partial h_1^{\ell_2}}(Y|(a_j^*)^\top X + b_j^*, \sigma_j^*)$$

$$+ \sum_{j=1}^{k_*} \sum_{|\gamma|=0}^{1+\mathbf{1}_{\{|\mathcal{A}_j|>1\}}} \omega_\gamma(j) \cdot X^\gamma \exp((\beta_{1j}^* + t_2)^\top X) g_{G_*}(Y|X) = 0,$$

for almost surely $(X, Y)$, where $\eta_{\ell_1,\ell_2}(j)$ and $\omega_\gamma(j)$ denote the limits of $T_{\ell_1,\ell_2}^n(j)/[m_n \mathcal{D}_2'(G_n, G_*)]$ and $S_\gamma^n(j)/[m_n \mathcal{D}_2'(G_n, G_*)]$ as $n \to \infty$, respectively, for all $j \in [k_*]$, $0 \leq 2|\ell_1| + \ell_2 \leq 2\bar{r}(|\mathcal{A}_j|)$ and $0 \leq |\gamma| \leq 1 + \mathbf{1}_{\{|\mathcal{A}_j|>1\}}$. By definition, at least one among $\eta_{\ell_1,\ell_2}(j)$ and $\omega_\gamma(j)$ is different from zero. Nevertheless, as the set

$$\mathcal{W}_2 := \left\{ X^{\ell_1} \exp((\beta_{1j}^* + t_2)^\top X) \frac{\partial^{\ell_2} f}{\partial h_1^{\ell_2}}(Y|(a_j^*)^\top X + b_j^*, \sigma_j^*) : j \in [k_*], \ 0 \leq 2|\ell_1| + \ell_2 \leq 2\bar{r}(|\mathcal{A}_j|) \right\}$$

$$\cup \left\{ X^\gamma \exp((\beta_{1j}^* + t_2)^\top X) g_{G_*}(Y|X) : j \in [k_*], \ 0 \leq |\gamma| \leq 1 + \mathbf{1}_{\{|\mathcal{A}_j|>1\}} \right\}, \tag{30}$$

is linearly independent w.r.t $X$ and $Y$ (proof can be done similarly to Lemma 2), it follows that

$$\eta_{\ell_1,\ell_2}(j) = \omega_\gamma(j) = 0,$$

for all $j \in [k_*]$, $0 \leq 2|\ell_1| + \ell_2 \leq 2\bar{r}(|\mathcal{A}_j|)$ and $0 \leq |\gamma| \leq 1 + \mathbf{1}_{\{|\mathcal{A}_j|>1\}}$, which is a contradiction. Hence, we achieve the inequality in equation (27), and complete the proof.

# B Proofs of Auxiliary Results

In this appendix, we provide proofs for the results of Proposition 1 and Proposition 2 in Appendix B.1 and Appendix B.2, respectively, while we leave that for Lemma 1 in Appendix B.3.

## B.1 Proof of Proposition 1

Given the notations in Proposition 1, assume that the equation $g_G(Y|X) = g_{G'}(Y|X)$ holds true, that is,

$$\sum_{i=1}^{k} \frac{\exp((\beta_{1i})^\top X + \beta_{0i})}{\sum_{j=1}^{k} \exp((\beta_{1j})^\top X + \beta_{0j})} \cdot f(Y|(a_i)^\top X + b_i, \sigma_i)$$

$$= \sum_{i=1}^{k'} \frac{\exp((\beta_{1i}')^\top X + \beta_{0i}')}{\sum_{j=1}^{k'} \exp((\beta_{1j}')^\top X + \beta_{0j}')} \cdot f(Y|(a_i')^\top X + b_i', \sigma_i'), \tag{31}$$

for almost surely $(X, Y)$. Then, it follows from the identifiability of the location-scale Gaussian mixtures [56, 57] that the number of components and the weight set of the mixing measure $G$ equal to those of its counterpart $G'$, i.e. $k = k'$ and

$$\left\{ \frac{\exp((\beta_{1i})^\top X + \beta_{0i})}{\sum_{j=1}^{k} \exp((\beta_{1j})^\top X + \beta_{0j})} : i \in [k] \right\} \equiv \left\{ \frac{\exp((\beta_{1i}')^\top X + \beta_{0i}')}{\sum_{j=1}^{k} \exp((\beta_{1j}')^\top X + \beta_{0j}')} : i \in [k] \right\},$$

for almost surely $X$. For simplicity, we may assume that

$$\frac{\exp((\beta_{1i})^\top X + \beta_{0i})}{\sum_{j=1}^{k} \exp((\beta_{1j})^\top X + \beta_{0j})} = \frac{\exp((\beta_{1i}')^\top X + \beta_{0i}')}{\sum_{j=1}^{k} \exp((\beta_{1j}')^\top X + \beta_{0j}')},$$

for all $i \in [k]$. Since the softmax function is invariant to translation, we get that $\beta_{0i} = \beta_{0i}' + t_1$ and $\beta_{1i} = \beta_{1i}' + t_2$ for some $t_1 \in \mathbb{R}$ and $t_2 \in \mathbb{R}^d$. Therefore, equation (31) reduces to

$$\sum_{i=1}^{k} \exp(\beta_{0i}) u(Y|X; \beta_{1i}, a_i, b_i, \sigma_i) = \sum_{i=1}^{k} \exp(\beta_{0i}) u(Y|X; \beta_{1i}, a_i', b_i', \sigma_i')), \tag{32}$$

for almost surely $(X, Y)$, where $u(Y|X; \beta_1, a, b, \sigma) := \exp(\beta_1^\top X) f(Y|a^\top X + b, \sigma)$ for all $i \in [k]$. Next, we will partition the index set $[k]$ into $q$ subsets $U_1, U_2, \ldots, U_q$ such that for each $\ell \in [q]$, we have $\exp(\beta_{0i}) = \exp(\beta_{0i'})$ for any $i, i' \in U_\ell$. As a result, equation (32) can be rewritten as

$$\sum_{\ell=1}^{q} \sum_{i \in U_\ell} \exp(\beta_{0i}) u(Y|X; \beta_{1i}, a_i, b_i, \sigma_i) = \sum_{\ell=1}^{q} \sum_{i \in U_\ell} \exp(\beta_{0i}) u(Y|X; \beta_{1i}, a_i', b_i', \sigma_i'),$$

for almost surely $(X, Y)$. Given the above equation, for each $\ell \in [q]$, we obtain that

$$\left\{ ((a_i)^\top X + b_i, \sigma_i) : i \in U_\ell \right\} \equiv \left\{ ((a_i')^\top X + b_i', \sigma_i') : i \in U_\ell \right\},$$

for almost surely $X$, which directly leads to

$$\{(a_i, b_i, \sigma_i) : i \in U_\ell\} \equiv \{(a_i', b_i', \sigma_i') : i \in U_\ell\}.$$

WLOG, we assume that $(a_i, b_i, \sigma_i) = (a_i', b_i', \sigma_i')$ for all $i \in U_\ell$. Consequently,

$$\sum_{\ell=1}^{q} \sum_{i \in U_\ell} \exp(\beta_{0i}) \delta_{\{\beta_{1i}, a_i, b_i, \sigma_i\}} = \sum_{\ell=1}^{q} \sum_{i \in U_\ell} \exp(\beta_{0i}' + t_1) \delta_{\{\beta_{1i}' + t_2, a_i', b_i', \sigma_i'\}},$$

or equivalently, $G \equiv G_{t_1, t_2}'$. Hence, the proof is completed.

## B.2 Proof of Proposition 2

Our proof will be based on the convergence rates of density estimation from MLE in Theorem 7.4 in [58]. Before stating this result here, let us introduce some necessary notations. Firstly, let $\mathcal{P}_k(\Theta)$ be the set of conditional densities of all mixing measures in $\mathcal{O}_k(\Theta)$, i.e., $\mathcal{P}_k(\Theta) := \{g_G(Y|X) : G \in \mathcal{O}_k(\Theta)\}$. Additionally, we define

$$\widetilde{\mathcal{P}}_k(\Theta) := \{g_{(G+G_*)/2}(Y|X) : G \in \mathcal{O}_k(\Theta)\},$$
$$\widetilde{\mathcal{P}}_k^{1/2}(\Theta) := \{g_{(G+G_*)/2}^{1/2}(Y|X) : G \in \mathcal{O}_k(\Theta)\}.$$

Next, for each $\delta > 0$, the Hellinger ball centered around the conditional density $g_{G_*}(Y|X)$ and intersected with the set $\widetilde{\mathcal{P}}_k^{1/2}(\Theta)$ is denoted by

$$\widetilde{\mathcal{P}}_k^{1/2}(\Theta, \delta) := \left\{ g^{1/2} \in \widetilde{\mathcal{P}}_k^{1/2}(\Theta) : h(g, g_{G_*}) \leq \delta \right\}.$$

Finally, in order to measure the size of the above set, [58] proposes using the following quantity:

$$\mathcal{J}_B(\delta, \widetilde{\mathcal{P}}_k^{1/2}(\Theta, \delta)) := \int_{\delta^2/2^{13}}^{\delta} H_B^{1/2}(t, \widetilde{\mathcal{P}}_k^{1/2}(\Theta, t), \|\cdot\|) \, \mathrm{d}t \vee \delta, \tag{33}$$

where $H_B(t, \widetilde{\mathcal{P}}_k^{1/2}(\Theta, t), \|\cdot\|)$ denotes the bracketing entropy [58] of $\widetilde{\mathcal{P}}_k^{1/2}(\Theta, u)$ under the 2-norm, and $t \vee \delta := \max\{t, \delta\}$. Now, we are ready to recall the statement of Theorem 7.4 in [58]:

**Theorem 3** (Theorem 7.4, [58]). *Take $\Psi(\delta) \geq \mathcal{J}_B(\delta, \widetilde{\mathcal{P}}_k^{1/2}(\Theta, \delta))$ that satisfies $\Psi(\delta)/\delta^2$ is a non-increasing function of $\delta$. Then, for some universal constant $c$ and for some sequence $(\delta_n)$ such that $\sqrt{n}\delta_n^2 \geq c\Psi(\delta_n)$, we achieve that*

$$\mathbb{P}\Big(\mathbb{E}_X[h(g_{\widehat{G}_n}(\cdot|X), g_{G_*}(\cdot|X))] > \delta\Big) \leq c \exp\left(-\frac{n\delta^2}{c^2}\right),$$

*for all $\delta \geq \delta_n$.*

The proof of this theorem can be seen in [58].

*Proof of Proposition 2.* Back to our main proof, since

$$H_B(t, \widetilde{\mathcal{P}}_k^{1/2}(\Theta, t), \|\cdot\|) \leq H_B(t, \mathcal{P}_k(\Theta, t), h)$$

for any $t > 0$, it follows from equation (33) that

$$\mathcal{J}_B(\delta, \widetilde{\mathcal{P}}_k^{1/2}(\Theta, \delta)) \leq \int_{\delta^2/2^{13}}^{\delta} H_B^{1/2}(t, \mathcal{P}_k(\Theta, t), h) \, \mathrm{d}t \vee \delta \lesssim \int_{\delta^2/2^{13}}^{\delta} \log(1/t) dt \vee \delta,$$

where we apply the upper bound of a bracketing entropy in Lemma 3 (cf. the end of this proof) in the second inequality. Let $\Psi(\delta) = \delta \cdot [\log(1/\delta)]^{1/2}$, we have $\Psi(\delta)/\delta^2$ is a non-increasing function of $\theta$. Moreover, the above equation deduces that $\Psi(\delta) \geq \mathcal{J}_B(\delta, \widetilde{\mathcal{P}}_k^{1/2}(\Theta, \delta))$. Additionally, let $\delta_n = \sqrt{\log(n)/n}$, we have that $\sqrt{n}\delta_n^2 \geq c\Psi(\delta_n)$ for some universal constant $c$. As all the assumptions are met, Theorem 3 gives us

$$\mathbb{P}(\mathbb{E}_X[h(g_{\widehat{G}_n}(\cdot|X), g_{G_*}(\cdot|X))] > C(\log(n)/n)^{1/2}) \lesssim \exp(-c\log(n)),$$

for some universal constant $C$ that depends only on $\Theta$. $\qquad\square$

For completion, we will provide the result regarding the upper bound of a bracketing entropy in the following lemma:

**Lemma 3.** *Assume that $\Theta$ is a bounded set, then the following inequality holds true for any $0 \leq \varepsilon \leq 1/2$:*

$$H_B(\varepsilon, \mathcal{P}_k(\Theta), h) \lesssim \log(1/\varepsilon).$$

*Proof of Lemma 3.* Firstly, we will establish an upper bound for the univariate Gaussian density $f(Y|a^\top X + b, \sigma)$. Since both $\mathcal{X}$ and $\Theta$ are bounded sets, there exist positive constants $\kappa, u, \ell$ such that $-\kappa \leq a^\top X + b \leq \kappa$ and $\ell \leq \sigma \leq u$. As a result,

$$f(Y|a^\top X + b, \sigma) = \frac{1}{\sqrt{2\pi h_2}} \exp\left(-\frac{(Y - h_1)^2}{2h_2}\right) \leq \frac{1}{\sqrt{2\pi\ell}}.$$

For any $|Y| \geq 2\kappa$, we have that $\frac{(Y-h_1)^2}{2h_2} \geq \frac{Y^2}{8u}$, which leads to

$$f(Y|a^\top X + b, \sigma) \leq \frac{1}{\sqrt{2\pi\ell}} \exp\left(-\frac{Y^2}{8u}\right).$$

Putting the above results together, we obtain that $f(Y|a^\top X + b, \sigma) \leq K(Y|X)$, where we define $K(Y|X) := \frac{1}{\sqrt{2\pi\ell}} \exp\left(-\frac{Y^2}{8u}\right)$ if $|Y| \geq 2\kappa$, and $K(Y|X) := \frac{1}{\sqrt{2\pi\ell}}$ otherwise.

Subsequently, let $\eta \leq \varepsilon$, we assume that the set $\mathcal{P}_k(\Theta)$ has an $\eta$-cover (under $\ell_1$-norm) denoted by $\{\pi_1, \ldots, \pi_N\}$, where $N := N(\eta, \mathcal{P}_k(\Theta), \|\cdot\|_1)$ is known as the $\eta$-covering number of $\mathcal{P}_k(\Theta)$. Then, we will build up the brackets of the form $[\nu_i(Y|X), \mu_i(Y|X)]$ for all $i \in [N]$ as follows:

$$\nu_i(Y|X) := \max\{\pi_i(Y|X) - \eta, 0\},$$
$$\mu_i(Y|X) := \max\{\pi_i(Y|X) + \eta, K(Y|X)\}.$$

Consequently, it can be checked that $\mathcal{P}_k(\Theta) \subset \bigcup_{i=1}^N [\nu_i(Y|X), \mu_i(Y|X)]$ with a note that $\mu_i(Y|X) - \nu_i(Y|X) \leq \min\{2\eta, K(Y|X)\}$. Next, for each $i \in [N]$, we attempt to give an upper bound for

$$\|\mu_i - \nu_i\|_1 = \int_{|Y| < 2\kappa} (\mu_i(Y|X) - \nu_i(Y|X)) \, \mathrm{d}(X, Y) + \int_{|Y| \geq 2\kappa} (\mu_i(Y|X) - \nu_i(Y|X)) \, \mathrm{d}(X, Y)$$

$$\leq R\eta + \exp\left(-\frac{R^2}{2u}\right) \leq R'\eta,$$

where $R := \max\{2\kappa, \sqrt{8u}\} \log(1/\eta)$ and $R'$ is some positive constant. By definition of the bracketing entropy, since $H_B(R'\eta, \mathcal{P}_k(\Theta), \|\cdot\|_1)$ is the logarithm of the smallest number of brackets of size $R'\eta$ necessary to cover $\mathcal{P}_k(\Theta)$, we achieve that

$$H_B(R'\eta, \mathcal{P}_k(\Theta), \|\cdot\|_1) \leq \log N = \log N(\eta, \mathcal{P}_k(\Theta), \|\cdot\|_1).$$

Assume that the following upper bound for the covering number $\log N(\eta, \mathcal{P}_k(\Theta), \|\cdot\|_1) \lesssim \log(1/\eta)$ holds true (proof is provided below), then the above result leads to

$$H_B(R'\eta, \mathcal{P}_k(\Theta), \|\cdot\|_1) \lesssim \log(1/\eta).$$

By selecting $\eta = \varepsilon/R'$, we receive that $H_B(\varepsilon, \mathcal{P}_k(\Theta), \|\cdot\|_1) \lesssim \log(1/\varepsilon)$. Furthermore, since the Hellinger distance is upper bounded by the $L^1$-norm, we reach the desired conclusion:

$$H_B(\varepsilon, \mathcal{P}_k(\Theta), h) \lesssim \log(1/\varepsilon).$$

**Upper bound of the covering number.** For completion, we will establish the following upper bound for the covering number, i.e.,

$$\log N(\eta, \mathcal{P}_k(\Theta), \|\cdot\|_1) \lesssim \log(1/\eta).$$

Let us denote $\Delta := \{(\beta_0, \beta_1) \in \mathbb{R} \times \mathbb{R}^d : (\beta_0, \beta_1, a, b, \sigma) \in \Theta\}$ and $\Omega := \{(a, b, \sigma) \in \mathbb{R}^d \times \mathbb{R} \times \mathbb{R}_+ : (\beta_0, \beta_1, a, b, \sigma) \in \Theta\}$. Since $\Theta$ is a compact set, $\Delta$ and $\Omega$ are also compact. Thus, we can find $\eta$-covers $\Delta_\eta$ and $\Omega_\eta$ for $\Delta$ and $\Omega$, respectively. It can be verified that $|\Delta_\eta| \leq \mathcal{O}(\eta^{-(d+1)k})$ and $|\Omega_\eta| \lesssim \mathcal{O}(\eta^{-(d+2)k})$.

For each mixing measure $G = \sum_{i=1}^k \exp(\beta_{0i})\delta_{(\beta_{1i}, a_i, b_i, \sigma_i)} \in \mathcal{O}_k(\Theta)$, we consider another one denoted by $\widetilde{G} := \sum_{i=1}^k \exp(\beta_{0i})\delta_{(\beta_{1i}, \overline{a}_i, \overline{b}_i, \overline{\sigma}_i)}$, where $(\overline{a}_i, \overline{b}_i, \overline{\sigma}_i) \in \Omega_\eta$ such that $(\overline{a}_i, \overline{b}_i, \overline{\sigma}_i)$ are the closest to $(a_i, b_i, \sigma_i)$ in that set for all $i \in [k]$. In addition, we also take into account the following mixing measure $\overline{G} := \sum_{i=1}^k \exp(\overline{\beta}_{0i})\delta_{(\overline{\beta}_{1i}, \overline{a}_i, \overline{b}_i, \overline{\sigma}_i)}$, where $(\overline{\beta}_{0i}, \overline{\beta}_{1i}) \in \Delta_\eta$ are the closest to $(\beta_{0i}, \beta_{1i})$ in that set. We can verify that the conditional density $g_{\overline{G}}$ belongs to the following set:

$$\mathcal{R} := \{g_G \in \mathcal{P}_k(\Theta) : (\beta_{0i}, \beta_{1i}) \in \Delta_\eta, \ (a_i, b_i, \sigma_i) \in \Omega_\eta, \ \forall i \in [k]\}.$$

Let us denote $\mathrm{Softmax}(\beta_{1i}^\top X + \beta_{0i}) := \dfrac{\exp(\beta_{1i}^\top X + \beta_{0i})}{\sum_{j=1}^k \exp(\beta_{1j}^\top X + \beta_{0j})}$. From the formulation of $\widetilde{G}$, we get the following bounds:

$$\|g_G - g_{\widetilde{G}}\|_1 \leq \sum_{i=1}^k \int_{\mathcal{X}} \mathrm{Softmax}(\beta_{1i}^\top X + \beta_{0i}) \Big| f(Y|(a_i)^\top X + b_i, \sigma_i) - f(Y|(\overline{a}_i)^\top X + \overline{b}_i, \overline{\sigma}_i) \Big| \mathrm{d}X$$

$$\leq \sum_{i=1}^k \int_{\mathcal{X}} \Big| f(Y|(a_i)^\top X + b_i, \sigma_i) - f(Y|(\overline{a}_i)^\top X + \overline{b}_i, \overline{\sigma}_i) \Big| \mathrm{d}X$$

$$\lesssim \sum_{i=1}^k (\|a_i - \overline{a}_i\| + |b_i - \overline{b}_i| + |\sigma_i - \overline{\sigma}_i|)$$

$$\lesssim \eta, \tag{34}$$

where the second inequality follows from the facts that $\mathcal{X}$ is a bounded set. Note that $\mathrm{Softmax}$ is a Lipschitz function with Lipschitz constant $L \geq 0$. Additionally, since $\mathcal{X}$ is a bounded set, there exists a constant $B > 0$ such that $\|X\| \leq B$ for any $X \in \mathcal{X}$. As a result, we get

$$\|g_{\widetilde{G}} - g_{\overline{G}}\|_1 \leq \sum_{i=1}^k \int_{\mathcal{X}} \Big| \mathrm{Softmax}(\beta_{1i}^\top X + \beta_{0i}) - \mathrm{Softmax}(\overline{\beta}_{1i}^\top X + \overline{\beta}_{0i}) \Big| f(Y|(\overline{a}_i)^\top X + \overline{b}_i, \overline{\sigma}_i) \mathrm{d}X$$

$$\lesssim L \sum_{i=1}^k \int_{\mathcal{X}} \Big( \|\beta_{1i} - \overline{\beta}_{1i}\| \cdot \|X\| + |\beta_{0i} - \overline{\beta}_{0i}| \Big) \mathrm{d}X$$

$$\leq Lk\eta(B+1), \tag{35}$$

where the second inequality follows from the fact that $\mathrm{Softmax}$ is a Lipschitz function and the Gaussian density $f(Y|(\overline{a}_i)^\top X + \overline{b}_i, \overline{\sigma}_i)$ is bounded. Putting the bounds in equations (34) and (35) together with the triangle inequality, we receive that

$$\|g_G - g_{\overline{G}}\|_1 \leq \|g_G - g_{\widetilde{G}}\|_1 + \|g_{\widetilde{G}} - g_{\overline{G}}\|_1 \lesssim \eta,$$

which means that $\mathcal{R}$ is an $\eta$-cover (not necessarily smallest) of the metric space $(\mathcal{P}_k(\Theta), \|\cdot\|_1)$. By definition of the covering number, we know that

$$N(\eta, \mathcal{P}_k(\Theta), \|\cdot\|_1) \leq |\mathcal{R}| = |\Delta_\eta| \times |\Omega_\eta| \leq \mathcal{O}(\eta^{-(d+1)k}) \cdot \mathcal{O}(\eta^{(-d+2)k}) \leq \mathcal{O}(\eta^{-(2d+3)k}),$$

which implies that,

$$\log N(\eta, \mathcal{P}_k(\Theta), \|\cdot\|_\infty) \lesssim \log(1/\eta).$$

Hence, the proof is completed. $\qquad\square$

## B.3 Proof of Lemma 1

First of all, let us recall the system of polynomial equations of interest here:

$$\sum_{j=1}^{m} \sum_{\alpha \in \mathcal{I}_{\ell_1,\ell_2}} \frac{p_{5j}^2 \, p_{1j}^{\alpha_1} \, p_{2j}^{\alpha_2} \, p_{3j}^{\alpha_3} \, p_{4j}^{\alpha_4}}{\alpha_1! \, \alpha_2! \, \alpha_3! \, \alpha_4!} = 0, \tag{36}$$

where $\mathcal{I}_{\ell_1,\ell_2} = \{\alpha = (\alpha_1, \alpha_2, \alpha_3, \alpha_4) \in \mathbb{N}^d \times \mathbb{N}^d \times \mathbb{N} \times \mathbb{N} : \alpha_1 + \alpha_2 = \ell_1, \, \alpha_3 + 2\alpha_4 = \ell_2 - |\alpha_2|\}$ for any $(\ell_1, \ell_2) \in \mathbb{N}^d \times \mathbb{N}$ such that $0 \le |\ell_1| \le r, 0 \le \ell_2 \le r - |\ell_1|$ and $|\ell_1| + \ell_2 \ge 1$.

In this proof, we denote $p_{1j} = (p_{1j1}, p_{1j2}, \ldots, p_{1jd})$ and $p_{2j} = (p_{2j1}, p_{2j2}, \ldots, p_{2jd})$.

**When $m = 2$:** By observing a portion of the above system when $\ell_1 = \mathbf{0}_d$, which is given by

$$\sum_{j=1}^{m} \sum_{\alpha_3 + 2\alpha_4 = \ell_2} \frac{p_{5j}^2 \, p_{3j}^{\alpha_3} \, p_{4j}^{\alpha_4}}{\alpha_3! \, \alpha_4!} = 0, \quad \ell_2 = 1, 2, \ldots, r. \tag{37}$$

It follows from Proposition 2.1 in [28] that the smallest natural number $r$ such that the system (37) does not have any non-trivial solutions when $m = 2$ is $r = 4$. It is worth noting that a solution of the system 37 is considered non-trivial in [28] if all the values of $p_{5j}$ are different from zero, whereas at least one among $p_{3j}$ is non-zero, which aligns with our definition of non-trivial solutions for the system (36). Thus, we get $\bar{r}(m) \le 4$, and it suffices to demonstrate that $\bar{r}(m) > 3$. Indeed, when $r = 3$, the system in equation (36) can be written as follows:

$$\sum_{j=1}^{m} p_{5j}^2 p_{1jl} = 0 \; \forall l \in [d], \quad \sum_{j=1}^{m} p_{5j}^2 p_{3j} = 0, \quad \sum_{j=1}^{m} p_{5j}^2 (p_{2ju} + p_{1jv} p_{3j}) = 0 \; \forall u, v \in [d],$$

$$\sum_{j=1}^{m} p_{5j}^2 p_{1ju} p_{1jv} = 0 \; \forall u, v \in [d], \quad \sum_{j=1}^{m} p_{5j}^2 \left( \frac{1}{2} p_{3j}^2 + p_{4j} \right) = 0, \quad \sum_{j=1}^{m} p_{5j}^2 \left( \frac{1}{3!} p_{3j}^3 + p_{3j} p_{4j} \right) = 0,$$

$$\sum_{j=1}^{m} p_{5j}^2 p_{1ju} p_{1jv} p_{1jl} = 0 \; \forall u, v, l \in [d], \quad \sum_{j=1}^{m} p_{5j}^2 \left( \frac{1}{2} p_{1ju} p_{1jv} p_{3j} + p_{1jl} p_{2j\tau} \right) = 0 \; \forall u, v, l, \tau \in [d],$$

$$\sum_{j=1}^{m} p_{5j}^2 \left( \frac{1}{2} p_{1ju} \cdot p_{3j}^2 + p_{1jv} p_{4j} + p_{2jl} p_{3j} \right) = 0 \; \forall u, v, l, \tau \in [d]. \tag{38}$$

It can be seen that the following is a non-trivial solution of the above system: $p_{5j} = 1, p_{1j} = p_{2j} = \mathbf{0}_d$ for all $j \in [m], p_{31} = 1, p_{32} = -1, p_{41} = p_{42} = -\frac{1}{2}$. Therefore, we obtain that $\bar{r}(m) > 3$, which leads to $\bar{r}(m) = 4$.

**When $m = 3$:** Note that $\bar{r}(m)$ is a monotonically increasing function of $m$. Therefore, it follows from the previous result that $\bar{r}(m) > \bar{r}(2) = 4$, or equivalently, $\bar{r}(m) \ge 5$. Additionally, according to Proposition 2.1 in [28], we deduce that $\bar{r}(m) \le 6$ based on the reduced system in equation (37). Thus, we only need to show that $\bar{r}(m) > 5$. Indeed, the system (36) when $r = 5$ is a combination of

the system in equation (38) and the following system:

$$\sum_{j=1}^{m} p_{5j}^2 p_{1ju}p_{1jv}p_{1jl}p_{1j\tau} = 0 \; \forall u,v,l,\tau \in [d], \quad \sum_{j=1}^{m} p_{5j}^2\Big(\frac{1}{4!}p_{3j}^4 + \frac{1}{2!}p_{3j}^2 p_{4j} + \frac{1}{2!}p_{4j}^2\Big) = 0,$$

$$\sum_{j=1}^{m} p_{5j}^2\Big(\frac{1}{3!}p_{1ju}p_{3j}^3 + p_{1jv}p_{3j}p_{4j} + \frac{1}{2!}p_{2jl}p_{3j}^2 + p_{2j\tau}p_{4j}\Big) = 0 \; \forall u,v,l,\tau \in [d],$$

$$\sum_{j=1}^{m} p_{5j}^2\Big(\frac{1}{3!}p_{1ju_1}p_{1ju_2}p_{1ju_3}p_{3j} + \frac{1}{2!}p_{1jv_1}p_{1jv_2}p_{2jv_3}\Big) = 0 \; \forall (u_i)_{i=1}^3, (v_i)_{i=1}^3 \in [d]^3,$$

$$\sum_{j=1}^{m} p_{5j}^2\Big(\frac{1}{2!2!}p_{1ju_1}p_{1ju_2}p_{3j}^2 + \frac{1}{2!}p_{1ju_3}p_{1ju_4}p_{4j} + p_{1ju_5}p_{1ju_6}p_{3j}\Big) = 0 \; \forall (u_i)_{i=1}^6 \in [d]^6,$$

$$\sum_{j=1}^{m} p_{5j}^2 \prod_{i=1}^{5} p_{1ju_i} = 0 \; \forall (u_i)_{i=1}^5 \in [d]^5,$$

$$\sum_{j=1}^{m} p_{5j}^2\Big(\frac{1}{5!}p_{3j}^5 + \frac{1}{3!}p_{3j}^3 p_{4j} + \frac{1}{2!}p_{3j}p_{4j}^2\Big) = 0,$$

$$\sum_{j=1}^{m} p_{5j}^2\Big(\frac{1}{4!}p_{1ju_1}p_{3j}^4 + \frac{1}{2!}p_{1ju_2}p_{3j}^2 p_{4j} + \frac{1}{2!}p_{1ju_3}p_{4j}^2 + \frac{1}{3!}p_{2ju_4}p_{3j}^3 + p_{2ju_5}p_{3j}p_{4j}\Big) = 0$$
$$\forall (u_i)_{i=1}^5 \in [d]^5,$$

$$\sum_{j=1}^{m} p_{5j}^2\Big(\frac{1}{4!}\prod_{i=1}^{4} p_{1ju_i}p_{3j} + \frac{1}{3!}\prod_{i=5}^{7} p_{1ju_i}p_{2ju_8}\Big) = 0 \; \forall (u_i)_{i=1}^8 \in [d]^8,$$

$$\sum_{j=1}^{m} p_{5j}^2\Big(\frac{1}{2!3!}\prod_{i=1}^{2} p_{1ju_i}p_{3j}^3 + \frac{1}{2!}\prod_{i=3}^{4} p_{1ju_i}p_{3j}p_{4j} + p_{1ju_5}p_{2ju_6}\big(\frac{1}{2}p_{3j}^2 + p_{4j}\big) + \frac{1}{2!}\prod_{i=7}^{8} p_{2ju_i}p_{3j}\Big) = 0$$
$$\forall (u_i)_{i=1}^8 \in [d]^8,$$

$$\sum_{j=1}^{m} p_{5j}^2\Big(\frac{1}{3!2!}\prod_{i=1}^{3} p_{1ju_i}p_{3j}^2 + \frac{1}{3!}\prod_{i=4}^{6} p_{1ju_i}p_{4j} + \frac{1}{2!}p_{1ju_7}p_{2ju_8}p_{3j} + \frac{1}{2!}p_{1ju_9}\prod_{i=10}^{11} p_{2ju_i}\Big) = 0$$
$$\forall (u_i)_{i=1}^{11} \in [d]^{11}.$$

We can verify that the following is a non-trivial solution of this system:

$$p_{5j} = 1, \quad p_{1j} = p_{2j} = \mathbf{0}_d, \quad \forall j \in [m],$$
$$p_{31} = \frac{\sqrt{3}}{3}, \quad p_{32} = -\frac{\sqrt{3}}{3}, \quad p_{33} = 0,$$
$$p_{41} = p_{42} = -\frac{1}{6}, \quad p_{43} = 0.$$

Hence, we conclude that $\bar{r}(m) = 6$.

## C  Experiments

In this appendix, we conduct a simulation study to empirically validate our theoretical results on the convergence rates of maximum likelihood estimation (MLE) in the softmax gating Gaussian mixture of experts established in Theorem 1 and Theorem 2.

### C.1  Numerical Schemes

We illustrate the heterogeneity convergence rates of the MLE under the softmax gating Gaussian mixture of experts via exact-fitted and over-fitted models, which correspond to the exact-fitted

and over-fitted settings described in Sections 3.1 and 3.2, respectively. For each case, we let $X$ be uniformly distributed over $[0, 1]$ and we generate observations $Y$ from the conditional density $g_{G_*}(Y|X)$ of softmax gating Gaussian mixture of experts model in equation (1). Here, the true mixing measure $G_* = \sum_{i=1}^{k_*} \exp(\beta_{0i}^*) \delta_{(\beta_{1i}^*, a_i^*, b_i^*, \sigma_i^*)}$, where $k_* = 2$, is specified as follows:

$$\beta_{01}^* = -8, \quad \beta_{11}^* = 25, \quad a_1^* = -20, \quad b_1^* = 15, \quad \sigma_1^* = 0.3,$$
$$\beta_{02}^* = 0, \quad \beta_{12}^* = 0, \quad a_2^* = 20, \quad b_2^* = -5, \quad \sigma_2^* = 0.4.$$

Then, we compute the MLE $\widehat{G}_n$ w.r.t. a number of components $k$ for each sample. For both of these settings, we choose $k \in \{k_*, k_* + 1, k_* + 2\}$. In order to perform the MLE, we use a numerical scheme based on the EM algorithm [12] similar to the one used by Chamroukhi et al. [7, 8]. Note that the main difference with a classical EM is in the maximization step, as there are no closed formulas for updating the softmax gating parameters $(\beta_{0i}, \beta_{1i}), \forall i \in [k]$. For this purpose, following the results of Chamroukhi et al. [7, 8], see also [38, 10, 23], we use a multi-class iterative reweighted least-squares algorithm. All code for our simulation study below was written in Python 3.9.13 on a standard Unix machine.

We choose the convergence criteria $\epsilon = 10^{-6}$ and 2000 maximum EM iterations. Our goal is to illustrate the theoretical properties of the estimator $\widehat{G}_n$. Therefore, we have initialized the EM algorithm in a favourable way. More specifically, we first randomly partitioned the set $\{1, \ldots, k\}$ into $k_*$ index sets $J_1, \ldots, J_{k_*}$, each containing at least one point, for any given $k$ and $k_*$ and for each replication. Finally, we sampled $\beta_{1j}^*$ (resp. $a_j^*, b_j^*, \sigma_j^*$) from a unique Gaussian distribution centered on $\beta_{1t}^*$ (resp. $a_t^*, b_t^*, \sigma_t^*$), with vanishing covariance so that $j \in J_t$.

## C.2  Empirical Convergence Rates

The empirical mean of discrepancies $\mathcal{D}_1$ and $\mathcal{D}_2$ between $\widehat{G}_n$ and $G_*$, and the choice of $k$ for exact-fitted and over-fitted models are reported in Figures 1 and 2, respectively. It can be observed that the average discrepancies from $\widehat{G}_n$ to $G_0$ vanish at a rate of $\mathcal{O}(n^{-1/2})$ up to a logarithmic factor, as envisaged by Theorems 1 and 2. Although these empirical rates of convergence are similar for the two models, they imply that the convergence behaviour of the individual fitted parameters is very different in an over-fitted setting, which was already discussed in more detail in the Section 3.2.

### C.2.1  Exact-fitted Model

We generate 40 samples of size $n$ for each setting, given 200 different choices of sample size $n$ between $10^2$ and $10^5$. The empirical parametric convergence rate of the MLE to $G_*$ under the metric $\mathcal{D}_1$ in Figure 1 is consistent with the theoretical rates of estimating the true parameters $\exp(\beta_{0i}^*), \beta_{1i}^*$ (up to translation), $a_i^*, b_i^*, \sigma_i^*$ for $i \in [k_*]$, which are of order $\mathcal{O}(n^{-1/2})$ up to logarithmic factors.

### C.2.2  Over-fitted Model

In the over-fitted setting, we generate 40 samples of size $n$ for each setting, given 200 different choices of sample size $n$ between $n_{\min} \approx 14 * 10^3$ for $k = k_* + 1$, $n_{\min} \approx 27 * 10^3$ for $k = k_* + 2$ and $n_{\max} = 10^5$. To the best of our knowledge, there is still a lack of theoretical understanding of EM performance, in particular an established algorithm that enjoys global convergence for the parameter estimation of the over-fitted softmax gating Gaussian mixture of experts. The most related theoretical results are only for the mixture of expert with covariate-free gating networks in [41, 40, 42]. This explains why in Figure 2 for over-fitted setting, we have not plotted the error bar due to the instability of the EM algorithm for finding the global solution. Moreover, the sample size must be large enough so that the empirical behaviour of the MLE from the EM algorithm matches the theoretical rate of order $\mathcal{O}(n^{-1/2})$ up to a logarithmic term.

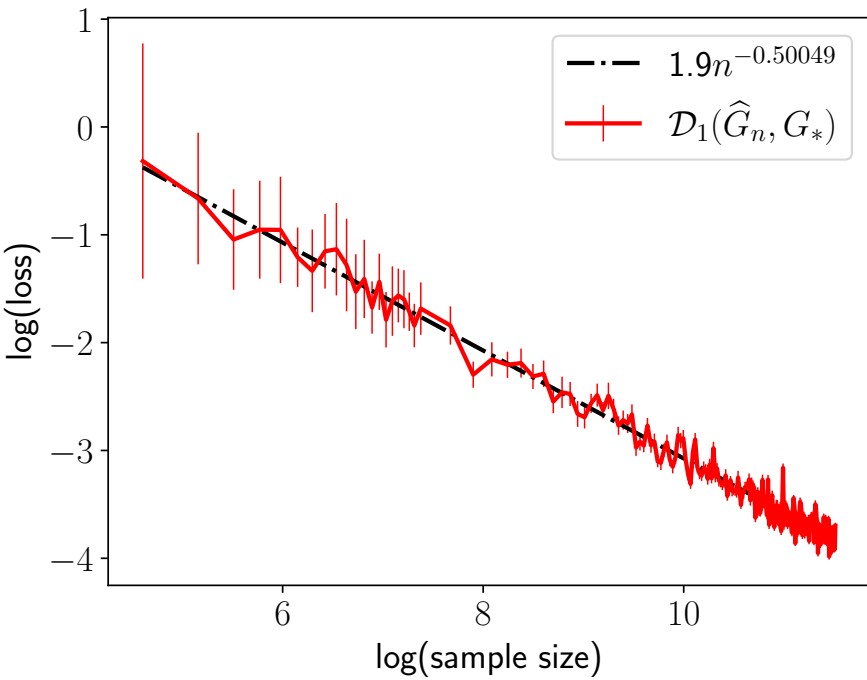

Figure 1: Log-log scaled plots of the simulation results for exact-fitted setting. We compute the estimator $\widehat{G}_n$ on 40 independent samples of size $n$ between $10^2$ and $10^5$. We plot its mean discrepancy from the true mixing measure in red, with error bars representing two empirical standard deviations. We also plot the least-squares fitted linear regression line of these points in a black dash-dotted line.

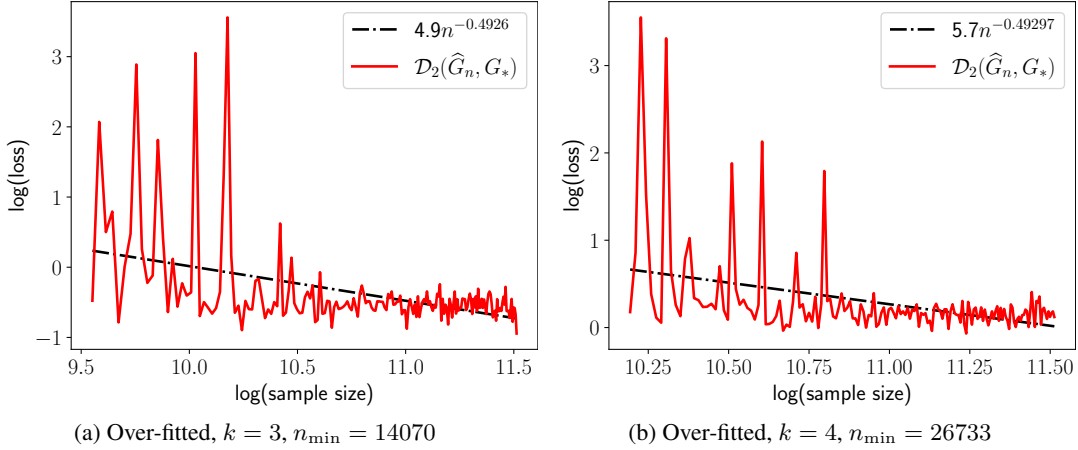

(a) Over-fitted, $k = 3$, $n_{\min} = 14070$

(b) Over-fitted, $k = 4$, $n_{\min} = 26733$

Figure 2: Log-log scaled plots of the empirical mean of the discrepancy $\mathcal{D}_2$ between $\widehat{G}_n$ and $G_*$ (red curves) and least-squares fitted linear regression (black dash-dotted lines) are shown using 40 independent sample sizes $n$ between $n_{\min}$ and $10^5$.

