# OpenReview forum: "Demystifying Softmax Gating Function in Gaussian Mixture of Experts"
_NeurIPS.cc/2023/Conference — NeurIPS 2023 spotlight_

### Official Review · Reviewer_8L63 · 2023-06-27

**Soundness:** 3 good
**Presentation:** 3 good
**Contribution:** 3 good
**Rating:** 7
**Confidence:** 2

**Summary:**

This paper addresses the long-standing challenge of parameter estimation in softmax gating Gaussian mixture of experts. The authors identify three theoretical challenges related to softmax gating: (i) the non-uniqueness of parameter estimation, (ii) the interaction between softmax gating and expert functions, and (iii) the complex dependence in the conditional density. They propose Vononoi loss functions to overcome these challenges and establish convergence rates for the maximum likelihood estimator (MLE). Additionally, they discover a connection between the MLE rate and a polynomial equation solvability problem when the number of experts is unknown or over-specified.

**Strengths:**

The paper proposes a novel solution to a long-standing problem in the literature, addressing three fundamental challenges associated with softmax gating. By introducing Vononoi loss functions and establishing convergence rates for the maximum likelihood estimator, the authors offer a fresh perspective on parameter estimation in these models. The paper contributes to the theoretical understanding of softmax gating Gaussian mixture of experts. It tackles the issues of non-uniqueness, interaction between gating and expert functions, and complex dependence in the conditional density. By resolving these challenges, the authors enhance our knowledge of the underlying principles and dynamics of these models. The paper establishes a connection between the MLE rate and a solvability problem of a system of polynomial equations. This connection adds an interesting aspect to the study, highlighting the potential relationships between statistical estimation and mathematical problem-solving.

**Weaknesses:**

Firstly, the absence of a comparison or discussion of popular techniques such as Gumbel-softmax [1] and max-propagate [2] raises concerns. These techniques are widely used in probabilistic MOE methods, and their exclusion from the paper may limit its completeness and applicability.

Secondly, the lack of experimental validation is another potential issue. The practical relevance of the proposed methods is claimed by the authors, but without empirical evidence or experiments, it is challenging to assess the true impact and effectiveness of the proposed approach in real-world scenarios.

[1] E. Jang, et al. "Categorical Reparameterization with Gumbel-Softmax." International Conference on Learning Representations (2017).

[2] J. Ren, et al. "Probabilistic mixture-of-experts for efficient deep reinforcement learning." arXiv preprint arXiv:2104.09122 (2021).

**Questions:**

No

---

> ### Author Rebuttal · Authors · 2023-08-08
>
> **Q1: The absence of a comparison or discussion of popular techniques such as Gumbel-softmax and max-propagate raises concerns. These techniques are widely used in probabilistic MOE methods, and their exclusion from the paper may limit its completeness and applicability.**
>
> Thanks for your suggestion, we will consider adding discussion about the Gumbel-Softmax and max propagation to the revision of our paper. However, we would like to emphasize that the main objective of this work is to study the convergence rates for parameter estimation in the Gaussian mixture of experts with softmax gating, which is commonly used in many machine learning and deep learning applications [1, 2] but remains missing theoretical understanding for long time. Therefore, the Gumbel-Softmax and max propagation techniques go beyond the scope of our paper, and we believe that proof techniques need to be further developed to handle these settings.
>
> **Q2: The lack of experimental validation is another potential issue, which makes it challenging to assess the true impact and effectiveness of the proposed approach in real-world scenarios.**
>
> Thanks for raising your concerns. Actually, in Appendix C of the supplementary material, we already carried out several experiments to empirically verify our theoretical results. Therefore, you can refer to that appendix for further details of experimental validation. Regarding the implication of theoretical results to real-world scenarios, please refer to the Common Question 2 in the General Response section.
>
> **References**
>
> [1] Hussein Hazimeh, Zhe Zhao, Aakanksha Chowdhery, Maheswaran Sathiamoorthy, Yihua Chen, Rahul Mazumder, Lichan Hong, and Ed H. Chi. Dselect-k: Differentiable selection in the mixture of experts with applications to multi-task learning. In Advances in Neural Information Processing Systems 34, 2021.
>
> [2] Noam Shazeer, Azalia Mirhoseini, Krzysztof Maziarz, Andy Davis, Quoc Le, Geoffrey Hinton, and Jeff Dean. 2017. Outrageously large neural networks: The sparsely-gated mixture-of-experts layer. In International Conference on Learning Representations 2017.

---

> > ### Comment · Reviewer_8L63 · 2023-08-11
> >
> > Thanks for the author's rebuttal! My concerns have been sufficiently addressed and I raised my score.

---

> > > ### Author Response · Authors · 2023-08-11
> > > **Thank You**
> > >
> > > We thank Reviewer 8L63 for your positive evaluation of our paper after the rebuttal and for increasing the score to accept.
> > >
> > > Best,
> > >
> > > The Authors

---

### Official Review · Reviewer_sK4C · 2023-07-02

**Soundness:** 3 good
**Presentation:** 4 excellent
**Contribution:** 3 good
**Rating:** 7
**Confidence:** 2

**Summary:**

The authors explore the theoretical complexity of standard Gaussian Mixture of Experts and propose a Vononoi loss which estimates different components of the MLE using different Voronoi cells.

Update: I have read the rebuttal and the other reviewers' responses, and will keep the score at 7 (accept).

**Strengths:**

The authors establish versions of the Voronoi loss in multiple scenarios, both in fixed settings and over-fitted settings.

The Voronoi loss function in both settings is computationally efficient up to a linear order of the number of experts, assuming that the true number of experts is fixed.

**Weaknesses:**

Unfortunately, there does not seem to be any practical exploration of the Voronoi loss in the real world setting to verify the theoretical findings, that should be fine due to the posit of the paper as a more theoretical work.

**Questions:**

Minor notes: there seems to be a switching between Vononoi vs Voronoi in the paper. Make sure to standardize them!

**Limitations:**

Yes.

---

> ### Author Rebuttal · Authors · 2023-08-08
>
> **Q1: Unfortunately, there does not seem to be any practical exploration of the Voronoi loss in the real world setting to verify the theoretical findings, that should be fine due to the posit of the paper as a more theoretical work.**
>
> Thanks for your comment. We would like to refer the reviewer to Appendix C in the supplementary material where we run several experiments under synthetic settings to empirically validate our theoretical results. For the connection of our theoretical results to real-world settings, please refer to the Common Question 2 in the General Response section.
>
>
> **Q2: Minor notes: there seems to be a switching between Vononoi vs Voronoi in the paper.**
>
> Thanks for pointing out, we will correct it in the revision of our paper.

---

### Official Review · Reviewer_txMC · 2023-07-20

**Soundness:** 4 excellent
**Presentation:** 3 good
**Contribution:** 2 fair
**Rating:** 7
**Confidence:** 2

**Summary:**

This paper establishes convergence rates for parameters (Gaussian mean / variance and expert weight) for softmax Gaussian mixture of experts. The rates depend on the setting: exact-fitted vs. over-fitted. The solutions are in terms of Voronoi losses and Hellinger distances.

**Strengths:**

**Originality**

To the best of my knowledge, this paper is the first to provide convergence rates for softmax Gaussian mixture of experts, making it an original paper. Likewise, the methodology for obtaining this result involves a rather involved setup, which struck me as highly non-trivial, again, supporting claims of originality. Of course, this is not my area of expertise, so it is difficult for me to accurately assess these points.

**Quality**

I believe the quality of the paper is strong. The authors provide a thorough description of the setup for softmax Gaussian mixture of experts, with rigorous proofs for the cases of exact-fitted and over-fitted number of experts. I have not assessed the quality of the proofs in the appendix.


**Clarity**

Despite tackling a complex theoretical problem and proposing a fairly involved solution, I found the paper was presented well. Terms are defined clearly, and the authors walk the reader through the main results before diving in to more detail.

**Significance**

The results seem marginally significant, as the authors claim that they have some practical implications for setting the number of experts and post-processing techniques, like merge-truncate-merge. However, fully demonstrating these practical insights, as well as empirical verification of the convergence rates, are not included in the paper, somewhat limiting their significance.

**Weaknesses:**

I have several concerns regarding the **significance** of the results.

First, these results target the convergence of the softmax mixture of expert parameters, however, in practical settings, I would imagine that these parameters are the outputs of deep networks, which may complicate the convergence picture.

Second, the convergence results are derived by considering the number of “true” experts, however, it’s unclear to me whether there is a notion of a “true” expert, particularly if the parameterization is used inside of a neural network / model to model a latent variable.

Finally, while I understand that this is an entirely theoretical paper, connecting it more closely to the practical settings in which softmax mixture of experts are used would help to more clearly demonstrate the significance of the paper. For instance, even if it’s in a toy setting, is there any way to empirically verify the convergence rates for exact-fitted vs. over-fitted setups?

Regarding **presentation**, I found the paper overall quite clear given the complexity of the result. However, some diagrams (there are none in the main paper) could possibly fill in any gaps in the reader’s understanding. For instance, diagrams showing the softmax Gaussian mixture of experts setting, as well as other concepts, like Voronoi cells, Hellinger distance, etc., would be useful additions to the paper.

**Questions:**

My main question is regarding the significance of the result and connecting the result to practical empirical settings — what can be done to make this connection clearer?

**Limitations:**

The authors have adequately addressed the main limitations of their work in the Discussion section of the paper.

---

> ### Author Rebuttal · Authors · 2023-08-08
>
> **Q1: The paper targets the convergence of the softmax mixture of expert parameters, however, in practical settings, I would imagine that these parameters are the outputs of deep networks, which may complicate the convergence picture.**
>
> Thanks for your question. Regarding the softmax gating Gaussian mixture of experts with general expert functions (including deep neural networks), the conditional density of $Y$ given $X$ is given by
> $$g_{G_*}(Y|X):=\sum_{i=1}^{k_*}\dfrac{\exp((\beta^*_{1i})^{\top}X+\beta^*_{0i})}{\sum_{j=1}^{k_*}\exp((\beta^*_{1j})^{\top}X+\beta^*_{0j})}f(Y|h_1(X,\theta^*_{1i}),h_2(X,\theta^*_{2i})),$$ where $f(\cdot|\mu,\sigma)$ denotes a Gaussian density with mean $\mu$ and variance $\sigma$, while $h_1:\mathcal{X}\times\Theta_1\to\Omega_1\subset\mathbb{R}$ and $h_2:\mathcal{X}\times\Theta_2\to\Omega_2\subset\mathbb{R}^+$ are referred to as expert functions which can be chosen as deep neural networks.
>
> We call that $h_1$ and $h_2$ are algebraically independent if they are twice differentiable w.r.t their parameters $\theta_1\in\Theta_1\subset\mathbb{R}^{q_1}$ and $\theta_2\in\Theta_2\subset\mathbb{R}^{q_2}$, and all the following sets are linearly independent w.r.t $X$:
>
> (S.1) $\{\dfrac{\partial h_1}{\partial\theta_1^{(u)}}(X,\theta_1)\dfrac{\partial h_1}{\partial\theta_1^{(v)}}(X,\theta_1), X^{w}\dfrac{\partial h_2}{\partial\theta_2^{(z)}}:1\leq u\leq v\leq q_1, 1\leq z\leq q_2,0\leq |w|\leq 1\}$,
>
> (S.2) $\{\dfrac{\partial h_2}{\partial\theta_2^{(u)}}(X,\theta_2)\dfrac{\partial h_2}{\partial\theta_2^{(v)}}(X,\theta_2):1\leq u\leq v\leq q_2\}$,
>
> (S.3) $\{\dfrac{\partial h_1}{\partial\theta_1^{(u)}}(X,\theta_1)\dfrac{\partial h_2}{\partial \theta_2^{(v)}}(X,\theta_2):1\leq u\leq q_1, 1\leq v\leq q_2\}$,
>
> (S.4) $\{X^{w}\dfrac{\partial h_1}{\partial\theta_1^{(u)}}(X,\theta_1):1\leq u\leq q_1,0\leq|w|\leq 1\}$,
>
> where $\theta_1^{(u)}$ are the $u$-th entry of $\theta_1$.
>
> For example, if $h_1(X,\theta_1)=\theta_1^{\top}X$ and $h_2(X,\theta_2)=\theta_2$, then $h_1$ and $h_2$ are algebraically independent.
>
> Under over-fitted settings, if $h_1$ and $h_2$ are algebraically independent expert functions, we propose using the follwoing Voronoi loss for the convergence analysis:
>
> $\mathcal{D}(G,G*):=\inf_{t_1,t_2}(\sum_{j:|\mathcal{A}_j|>1,i\in\mathcal{A}_j}[$
>
> $\exp(\beta_{0i})||(\Delta_{t_2}\beta_{1ij},\Delta\theta_{1ij},\Delta\theta_{2ij})||^2]$
>
> $+\sum_{j:|\mathcal{A}_j|=1,i\in\mathcal{A}_j}[$
>
> $\exp(\beta_{0i})||(\Delta_{t_2}\beta_{1ij},\Delta\theta_{1ij},\Delta\theta_{2ij})||]$
>
> $+\sum_{1\leq j\leq k_*}[$
>
> $|\sum_{i\in\mathcal{A}_j}[$
>
> $\exp(\beta_{0i})]-\exp(\beta^*_{0i}+t_1)|])$,
>
> where $\Delta_{t_2}\beta_{1ij}:=\beta_{1i}-(\beta^*_{1j}+t_2)$, $\Delta\theta_{1ij}=\theta_{1i}-\theta^*_{1j}$, $\Delta\theta_{2ij}=\theta_{2i}-\theta_{2j}$.
>
> **(we apologize for the typing of the loss function as it is too complex to compile in one line)**
>
> In particular, we find out that the MLE $\widehat{G}_n$ defined as in our work still converges to
>
> the true mixing measure $G_*$ under the above loss at a rate of $\mathcal{O}(n^{-1/2})$ (up to some logarithmic term).
> Consequently, the estimation rates for over-fitted parameters $\beta^*_{1j},\theta^*_{1j},\theta^*_{2j}$ are $\mathcal{O}(n^{-1/4})$, while those for exact-fitted parameters are $\mathcal{O}(n^{-1/2})$.
>
> However, when $h_1$ and $h_2$ are not algebraically independent, then the estimation rates for true parameters will vary with the choices of expert functions. For instance, in our work, $h_1(X,(a,b))=a^{\top}X+b$ and $h_2(X,\sigma)=\sigma$ are not algebraically independent since the corresponding set (S.4) is linearly dependent.
>
> **Q2: The convergence results are derived by considering the number of “true” experts, however, it’s unclear to me whether there is a notion of a “true” expert.**
>
> Thanks for your question. We would like to clarify that we did not use the term ``the number of true experts'' in our paper. However, we used two other terms which possibly related to that term:
>
> 1) ``the true number of experts'' (line 118), which was denoted by $k_*$;
>
> 2) ``true experts'' (first line on page 6), which indicates the experts with true parameters, e.g. $f(Y|(a^*_i)^{\top}X+b^*_i,\sigma^*_i)$.
>
> **Q3: Is there any way to empirically verify the convergence rates for exact-fitted vs. over-fitted setups?**
>
> Thanks for your question. Ones can refer to Appendix C in the supplementary material where we conducted several experiments to empirically justify the theoretical convergence rates for both exact-fitted and over-fitted settings.
>
> **Q4: Some diagrams could possibly fill in any gaps in the reader’s understanding. For instance, diagrams showing the softmax Gaussian mixture of experts setting, as well as other concepts, like Voronoi cells, Hellinger distance, etc.**
>
> Thanks for your suggestion, we will add the illustration of Voronoi cells in the PDF file (attached at the end of the General Response section) to the revision of our paper.
>
> Image Caption: Illustration of Voronoi cells under the over-fitted settings. Blue triangles represent true components while red rounds indicates fitted components. By definition, each Voronoi cell corresponds to only one true component, i.e. one blue triangle, and its cardinality is the number of corresponding fitted components, i.e. the number of red rounds. For instance, in cell 1, the blue triangle is approximated by two rounds, implying that the cardinality of cell 1 is two. Then, from the comments on Theorem 2 (line 237-244), since $\bar{r}(2)=4$, the estimation rates for $\beta^*_{11},b^*_1$ are $\mathcal{O}(n^{-1/8})$, while those for $a^*_1,\sigma^*_1$ are $\mathcal{O}(n^{-1/4})$ (up to some logarithmic term).
>
> **Q5: Regarding the significance of the result and connecting the result to practical empirical settings — what can be done to make this connection clearer?**
>
> Thanks for your question. Please refer to the Common Question 2 in the General Response section for our response.

---

> > ### Comment · Reviewer_txMC · 2023-08-15
> > **Response to Authors**
> >
> > Thank you for your rebuttal. After reading the other reviews and the authors' responses, I have decided to raise my score.

---

> > > ### Author Response · Authors · 2023-08-15
> > >
> > > We thank Reviewer txMC for your positive evaluation of our paper after the rebuttal and for increasing the score to accept (7). We really appreciate that!
> > >
> > > Best,
> > >
> > > The Authors

---

### Official Review · Reviewer_q5b3 · 2023-07-25

**Soundness:** 3 good
**Presentation:** 2 fair
**Contribution:** 3 good
**Rating:** 6
**Confidence:** 2

**Summary:**

This paper studies convergence rate of Maximum Likelihood Estimate (MLE) of parameters of Gaussian mixture of experts with softmax gating. Unlike some of the previous papers, the authors allow for the gating probabilities to be input dependent. Convergence rate (upper bound) is provided for both the setting where the number of components is known, and when only a lower bound is known. For the latter case, the authors show connection between the MLE convergence rate and solvability of a central system of polynomial equations.

**Strengths:**

* Makes progress on a difficult problem complicated by lack of identifiability and complex interactions between parameters of the gating mechanism and of the individual components.

* Proposes a new "Voronoi loss", which lower bound the Hellinger distance, and is crafted to capture properties of the mixture of experts model (the involved Voronoi cells roughly correspond to the subsets of the input space dominated by a particular expert under the ground truth).

* AFAICT provides the first MLE rates for models where the gating mixture is input dependent.

**Weaknesses:**

* As someone with only cursory understanding of the area (emergency reviewer), the paper was hard to follow and full of technical detail that to me felt like obfuscating rather than facilitating understanding.

* Unclear what can a practitioner take away, except that they should set the number of components as low as possible. (This is a very minor weakness, as I don't think every paper needs to have direct application.)

**Questions:**

* With the problem of identifiability, what happens if we introduce some penalty ($\ell^1$, $\ell^2$)? I was surprised that this (perhaps naive) solution was not discussed in the background section or anywhere else.

* Could you make more explicit how the level of separation between individual components of the ground truth impacts the convergence rate? Intuitively, the more separated, the easier the estimation problem should be.

* Potentially related question: What do you mean when you say that a particular Voronoi cell contains only one component of the MLE? Isn't the MLE a random variable? (Wouldn't that imply that the identity of cells which have only one component changes? Also what about the cells which have zero MLE components in them?)

---

> ### Author Rebuttal · Authors · 2023-08-08
>
> **Q1: As someone with only cursory understanding of the area, the paper was hard to follow and full of technical detail that to me felt like obfuscating rather than facilitating understanding.**
>
> Thanks for your comment. We would like to refer you to Common Question 1 in the General Response to understand our paper better.
>
> **Q2: Minor issue: Unclear what can a practitioner take away, except that they should set the number of components as low as possible.**
>
> Thank you for raising your concerns. In fact, there are two other important messages: (1) The slow convergence rates of the MLE may provide important thresholds for the merge-truncate-merge procedure, a procedure that has been used to estimate the true number of components in standard mixture models [1], to consistently estimate the true number of experts $k_*$ in our mixture of experts; (2) We also illustrate the heterogeneity convergence rates in the MLE for solving the parameter estimation of the softmax gating Gaussian mixture of experts via exact-fitted and over-fitted models. Although these empirical rates of convergence are similar for the two models, they imply that the convergence behaviour of the individual fitted parameters is very different in an over-fitted setting, which is already discussed in more detail in the Section 3.2 of our paper.
>
> **Q3: With the problem of identifiability, what happens if we introduce some penalty $(\ell_1,\ell_2)$?**
>
> Thank you for your question. As far as we know, we are not sure if introducing a penalty $(\ell_1,\ell_2)$ will solve the identifiability problem. Furthermore, proof techniques should be further developed to extend our proposed approach to these settings in order to establish the convergence rates for the parameter estimation problem in penalized mixture of experts.
>
> **Q4: Could you make more explicit how the level of separation between individual components of the ground truth impacts the convergence rate? Intuitively, the more separated, the easier the estimation problem should be.**
>
> Thanks for your question. In this work, we do not take into account any assumptions on the level of separation between ground-truth parameters. Moreover, it can be seen from Theorem 1 and Theorem 2 that the estimation rates for ground-truth parameters are independent of that separation level. Instead, under the over-fitted settings, those estimation rates depend on the solvability problem of a system of polynomial equations. For instance, Theorem 2 implies that a true parameter $a^*_j$ has an estimation rate of order
>
> $\mathcal{O}(n^{-1/\bar{r}(|\mathcal{A}_j|)})$ (up to some logarithmic term)
>
> for any $j\in\{1,2,\ldots,k_*\}$.
>
> Since $\bar{r}$ is a monotonically increasing function of $|\mathcal{A}_j|$, then the lower the cardinality of the Voronoi cell $\mathcal{A}_j$ is, the better the previous estimation rate for $a^*_j$ becomes.
>
> **Q5: What do you mean when you say that a particular Voronoi cell contains only one component of the MLE? Isn't the MLE a random variable? (Wouldn't that imply that the identity of cells which have only one component changes? Also what about the cells which have zero MLE components in them?)**
>
> Thanks for your question. We would like to emphasize that as the MLE $\widehat{G}_n$ is a random variable,
>
> then the Voronoi cells $\mathcal{A}^n_j$ for $j\in\{1,2,\ldots,k_*\}$ corresponding to the MLE are also random.
>
> Under the over-fitted settings, the number of components of the MLE $\widehat{G}_n$
>
> is strictly larger than that of the true mixing measure $G_*$, i.e. $\widehat{k}_{n}>k*$.
>
> Thus, according to the pigeonhole principle, there must exist some Voronoi cell having at least two elements. Additionally, we could also find some Voronoi cell $\mathcal{A}^n_j$ which has only one element $i$, and this element corresponds to the component $(\widehat{\beta}_{1i},\widehat{a}_i,\widehat{b}_i,\widehat{\sigma}_i)$ of the MLE. This is what we mean when we mention a Voronoi cell which contains only one component of the MLE.
>
> Regarding the question about Voronoi cells having zero MLE component, we get from Theorem 2 that the Voronoi loss $\mathcal{D}_2(\widehat{G}_n,G*)$ converges to zero as $n$ goes to infinity.
>
> Then, it follows from the definition of this Voronoi loss that $(\widehat{\beta}_{1i},\widehat{a}_i,\widehat{b}_i,\widehat{\sigma}_i)$ converges to
>
> $\(\beta^*_{1j},a^*_{j},b^*_{j},\sigma^*_{j})$ for any $i\in\mathcal{A}^n_j$ and $j\in\{1,2,\ldots,k_*\}$. This result implies that each Voronoi cell $\mathcal{A}^n_j$ must have at least one element.
>
> **References**
>
> [1] A. Guha, N. Ho, and X. Nguyen. On posterior contraction of parameters and interpretability in Bayesian mixture modeling. Bernoulli, 27(4):2159–2188, 2021.

---

### Official Review · Reviewer_PaPA · 2023-07-27

**Soundness:** 3 good
**Presentation:** 2 fair
**Contribution:** 3 good
**Rating:** 7
**Confidence:** 3

**Summary:**

This work considers the problem of parameter estimation for the **softmax-gated mixture of Gaussian experts**. For simplicity, first consider the case in which the mixture weights do not depend on the covariates X. Then, this model can be seen as a mixture of linear regressions where each “regression component” is corrupted by Gaussian noise. The paper considers the much more challenging case where the mixture weights depend on the covariates. The mixing weights are modeled as linear functions which are normalized by applying softmax to the vector of mixing weights.

The paper proposes a novel "Voronoi distance" for the parameter space, which takes into account the invariances of the parametric model, and is used to determine convergence rates for the maximum likelihood estimator (MLE). In the well-specified case where $k$, the number of components of the model used, is equal to $k^*$, the true number of components, the MLE parameters converge at the parametric rate $n^{-1/2}$. In the overparametrized case where $k > k^*$, the rates of the component parameters are slower and depend on how many components of the overparametrized model fall inside the Voronoi cell of the true model’s components.

**Strengths:**

- The paper overcomes significant technical challenges that come from using softmax gates for the mixing weights. The main contribution of the paper is showing that the expected total variation distance of the MLE’s conditional distribution $\hat{p}(Y|X)$ from the true conditional distribution $p(Y|X)$ is *lower bounded* by the proposed Voronoi distance. The Voronoi distance takes care of the inherent symmetries of the parameter space. For example, the mixing weight parameters can only be identified up to global translations due to the softmax operation.
- Since one can show that the expected TV distance of the MLE converges at the parametric rate $n^{-1/2}$ by computing the bracketing number of the parametric family [van de Geer, 2000; Theorem 7.2], this leads to an upper bound on the Voronoi distance and thus the $\ell_2$ distance of the MLE parameters.
- To be more precise, the paper proposes a family of Voronoi distances, for both the well-specified case and overparameterized case. The convergence rates for overparametrized cases are determined implicitly by a system of polynomial equations which in turn come from analyzing the Taylor expansion of the model’s conditional density w.r.t. the parameters.

**Weaknesses:**

**Exposition.** Though the technical contributions of this paper are solid, its exposition could be improved to make it more accessible to non-expert readers.

- **System of polynomial equations (Section 1).** The full details of the system of polynomial equations in the introduction is difficult to follow and seems unnecessary since it is reproduced in Section 3. It would be easier and less intimidating for the readers if the authors omitted the full details and simply noted that $r(m)$ is related to whether some system of polynomial equations admits no non-trivial solutions. Then, one could end with “we conjecture r(m) = 2m, and establish this for the first few cases m=1,2,3, in Lemma 1.” It would also be great if the authors could provide more explanation for how one arrives at these polynomial equations.
- **Computation of the Voronoi loss function (Section 3.2).** This would fit better in the experiment section. In typical estimation settings, we do not have access to the true parameters anyway. This quantity is only computable in synthetic settings where the experimenter knows the true parameters.
- **Practical implication paragraphs.** These paragraphs in Section 1 and Section 3.2 are redundant.

**Questions:**

- The authors mention in Section 1 that understanding the parameter estimation problem for mixture of experts models has been a long-standing open problem. Could the authors provide explanation for why this was the case? What were the technical barriers and what recent technical advances or breakthroughs have paved the way for the results established here?
- Which aspects of the Voronoi distances are novel? Assigning the model’s components according to the Voronoi cells of true components seems like a natural thing to do for any mixture model. Is it the \inf over translations of the mixing weights? Or the rate functions r(m) related to the system of polynomial equations?

**Editorial comment**

- In the introduction, it would motivate the paper’s result more if the authors provided simple and practical examples showing the importance of parameter estimation in this particular mixture of experts model rather than just estimating the density. Perhaps there are examples in which the parameters in the mixture of experts model are interpretable and have practical value?
- Typo. First paragraph of Section 3.3. Total variation distance should be V not h.
- It would be helpful to explain what mixing measures when they are first introduced in Section 1. Perhaps after Eq.(1) in the **Problem setting** paragraph.

---

> ### Author Rebuttal · Authors · 2023-08-08
>
> **Q1: The full details of the system of polynomial equations in the introduction are unnecessary and should be omitted. Instead, the authors should briefly introduce the quantity $r(m)$ and its properties. Additionally, how do the authors arrive at these polynomial equations?**
>
> Thanks for your suggestion, we will consider it and edit our paper accordingly. Regarding the derivation of the system of polynomial equations, please refer to parts (ii) and (iii) of the Common Question 1 in the General Response section for brief explanation. For further details, ones can refer to Step 2 of the proof of Theorem 2 from line 629 -649 in the supplementary material for further details.
>
> **Q2: Regarding the computation of the Voronoi loss functions, since we do not have access to the true parameters in typical estimation settings, these functions are only computable in synthetic settings where we know true parameters.**
>
> We agree with the reviewer that our proposed Voronoi loss functions are only computable when ones have access to true parameters. However, we would like to emphasize that the important feature of the proposed Voronoi losses is its ability to provide exact convergence rates of individual parameters of the MLE, i.e., we know the separation among the estimated parameters based on their distances to the true parameter or how small the weights of certain experts to 0 (see our response to Q5 for more details). From the practical perspective, these characterizations of separation among estimated parameters or the values of weights of experts give insight into the thresholds to merge these parameters when they become too close or truncate certain experts when their weights are too small. This procedure yields a useful method to reduce the complexity of the Gaussian mixture of experts and consistently determines the correct number of experts when the sample size is sufficiently large.
>
> **Q3: The practical implication paragraphs in Section 1 and Section 3.2 are redundant.**
>
> Thanks for your comment, we will consider removing the practical implication paragraph in Section 3.2 in the revision of our paper. However, we may keep the one in Section 1 as it provides an insight of how to choose the number of experts to achieve the best possible convergence rates of parameter estimation and connection to the merge-truncate-merge procedure, which is used to estimate the true number of experts.
>
> **Q4: Could the authors explain why understanding the parameter estimation problem for mixture of experts models has been a long-standing open problem? What were the technical barriers and what recent technical advances or breakthroughs have paved the way for the results established here?**
>
> Thanks for your questions. We would like to refer you to Common Question 1 in the General Response for further details.
>
> **Q5: Which aspects of the Voronoi distances are novel? Is it the $\inf$ over translations of the mixing weights? Or the rate functions $r(m)$ related to the system of polynomial equations?**
>
> Thanks for your question. Our proposed Voronoi loss functions are considered novel since they are able to characterize accurately the estimation rate for each parameter. In particular, from our Theorem 2, we know that the Voronoi loss $\mathcal{D}_{2}(\widehat{G}_n,G*)$ converges to zero at a rate of order $\mathcal{O}(n^{-1/2})$ (up to some logarithmic term).
>
> Then, it follows from the formulation of this loss that for each $j=1,2,\ldots,k_*$,
>
> all the resulting estimation rates for parameters $\beta^*_{1j},a^*_{j},b^*_{j},\sigma^*_{j}$ depend on the quantity $\bar{r}(|\mathcal{A}_j|)$, which possibly admits distinct values for different indices $j$. More specifically, if two Voronoi cells $\mathcal{A}_1$ and $\mathcal{A}_2$ does not share the same cardinality, i.e. $|\mathcal{A}_1|\neq|\mathcal{A}_2|$,
>
> then the estimation rates for $\beta^*_{11},a^*_1,b^*_1,\sigma^*_1$ are different from those of
>
> $\beta^*_{12},a^*_2,b^*_2,\sigma^*_2$, respectively.
>
> By contrast, if we adopted the Wasserstein loss as in [1], the estimation rate for
>
> $\beta^*_{1j}$ (resp. $a^*_j,b^*_j,\sigma^*_j$)
>
> remains unchanged for any $j=1,2,\ldots,k_*$, which is not accurate.
>
> **Q6: In the introduction, the authors should provide simple and practical examples showing the importance of parameter estimation in this particular mixture of experts model.**
>
> Thanks for your suggestion. Mixture models and mixture of experts are well-known for modelling heterogeneous data, and they are commonly used in medicine [2] and physical sciences [3]. In these applications, mixture parameters play a vital role in capturing the heterogeneity of the data, thus, the main objective in [2, 3] is to conduct statistical inference for those parameters. This leads to a need for determining the convergence rates for parameter estimation in mixture models and mixture of experts. We will add this discussion to the revision of our paper.
>
> **Q7: Typo. First paragraph of Section 3.3. Total variation distance should be $V$ not $h$.**
>
> Thanks for pointing out, we will correct it in the revision of our paper.
>
> **Q8: It would be helpful to explain what mixing measures are when they are first introduced in Section 1.**
>
> Thanks for your suggestion, we will add the definition of a mixing measure to the revision of our paper. Basically, a mixing measure is referred to as a mixture of some measures. For instance, in line 53 of our work, since $G_*$ is defined as a mixture of Dirac measures, it is called a mixing measure.
>
> **References**
>
> [1] N. Ho, C.-Y. Yang, and M. I. Jordan. Convergence rates for Gaussian mixtures of experts. Journal of Machine Learning Research, 2022.
>
> [2] Q. Li, R. Shi, and F. Liang. Drug sensitivity prediction with high-dimensional mixture regression. PloS one, 2019.
>
> [3] Kuusela, M. Semi-supervised anomaly detection—towards model-independent searches of new physics. In Journal of Physics: Conference Series, volume 368, 2012.

---

> > ### Comment · Reviewer_PaPA · 2023-08-14
> >
> > Thank you for your thorough response and clarification!

---

> > > ### Author Response · Authors · 2023-08-15
> > >
> > > We thank Reviewer PaPA for your positive evaluation of our paper after the rebuttal and for keeping your score of 7.
> > >
> > > Best,
> > >
> > > The Authors

---

### Author Rebuttal · Authors · 2023-08-08

**General Response**

Dear AC and reviewers,

Thanks for your thoughtful reviews and valuable comments, which have helped us enhance our paper substantially. In this section, we will address some common concerns from the reviewers, and then include these changes to the revision of our paper.

**CQ1: Main challenges in the paper?**

To the best of our knowledge, previous work on comprehending the convergence rates for parameter estimation problem in mixture of experts or mixture models [1] all assume that the corresponding gating or mixing proportion is independent of the covariates $X$. This assumption is limited and far from recent applications of mixture of experts in machine learning and deep learning [2], which leverage covariate-dependent gating such as softmax gating and its variants. For those reasons, understanding parameter estimation of softmax gating Gaussian mixture of experts has remained a long-standing open problem in the literature.

There are three main challenges in our work:

**(i)** Firstly, parameters $\beta^*_{1i}, \beta^*_{0i}$ of the softmax gating are not identifiable as those of the covariate-independent gating in previous work. Instead, they are identifiable up to translation, that is, the softmax gating does not change when we translate the parameters as follows: $\beta^*_{1i}+t_1$ and $\beta^*_{0i}+t_2$. As a consequence, we need to introduce an infimum operator in the Voronoi loss function to deal with this issue.

**(ii)** Secondly, it is clear that the numerators and denominators of softmax gating are dependent. Thus, in Step 1 of our proofs, if we applied the Taylor expansion directly to the conditional density difference $g_{G_n}(Y|X)-g_{G_*}(Y|X)$ as in previous work [1], we were unable to represent the conditional density difference as a linear combination of elements belonging to some linearly independent set, which is a key step in the proof techniques. To this end, we consider the product of the softmax gating's denominator and the conditional density difference, which is denoted by $Q_n$. Subsequently, we decompose the product $Q_n$ such that the decomposition includes two functions $\exp(\beta_{1i}^{\top}X)f(Y|a_i^{\top}X+b_i,\sigma_i)$ and $\exp(\beta_{1i}^{\top}X)g_{G_n}(Y|X)$ as in line 291. Then, we have to apply two Taylor expansions of different orders to these functions, respectively, rather than only one Taylor expansion as in [1]. Now, $Q_n$ is written as a linear combination but there are some linearly dependent terms due to the intrinsic interaction between the softmax gating's numerator $\exp(\beta_{1i}^{\top}X+\beta_{0i})$ and the Gaussian density function $f(Y|a_i^{\top}X+b_i,\sigma_i)$ via the partial differential equation in Eq.(3): $$\frac{\partial u(X,Y)}{\partial\beta_1}\cdot\frac{\partial u(X,Y)}{\partial b}=\frac{\partial u(X,Y)}{\partial\beta_0}\cdot\frac{\partial u(X,Y)}{\partial a},$$ where $u(X,Y):=\exp(\beta_1^{\top}X+\beta_0)f(Y|a^{\top}X+b,\sigma)$. Therefore, it takes us much effort to group these linearly dependent terms together as in Eq.(25) to formulate $Q_n$ as a linear combination of linearly independent terms.

**(iii)** Finally, in Step 2 of our proofs, we assume that all the ratios of coefficients in the above linear combination to the Voronoi loss converge to zero. Then, via some transformations, those limits lead to the system of polynomial equations in Eq.(6). Compared to the system in previous work [Eq.(6), 1], our system is much more complex and challenging due to the interaction between the softmax gating's numerator and the Gaussian density. As a result, it takes us greater effort to comprehend our system in Lemma 1.

**CQ2: Connection between theoretical results and real-world settings?**

There are two different real-world settings that the current theoretical results will yield important insights into:

(i) Well-specified setting: In this setting, we assume that the data are generated from Gaussian mixture of experts with softmax gating. It is the setting that we mainly study in the current work. Our results in the paper suggest that when we overspecify the number of experts in the well-specified setting, the convergence rates of some parameters can be very slow and depend on the amount of components that we overspecify the model while those of the remaining parameters can be very fast. These rates can be captured precisely by the novel Voronoi losses. The precise rates are important for developing model selection to choose the correct number of experts in the well-specified settings, which is important to improve the complexity of using Gaussian mixture of experts in practice (Please refer to the Practical Implication paragraph in the Introduction).

(ii) Misspecified setting: In this setting, the data are not generated from Gaussian mixture of experts with softmax gating while we fit the data by Gaussian mixture of experts. Different from the well-specified setting, the MLE $\widehat{G}_{n}$ converges to the mixing measures $\bar{G}$ which is an optimal solution for the problem of minimizing the following Kullback-Leibler divergence:

$KL(g_{G}(Y|X), P(Y|X))$,

where $P(Y|X)$ is the true conditional density function of $Y|X$, which is not a Gaussian mixture of experts, $g_{G}(Y|X)$ is Gaussian mixture of experts with mixing measure $G$. The insights from our theories in the well-specified setting indicate that the Voronoi losses can be used to obtain the precise rates of individual parameters of the MLE $\widehat{G}_{n}$ to those of $\bar{G}$. The detailed theoretical development of parameter estimation under the misspecified setting is beyond the current scope of the paper and left for future work.

**References**

[1] N. Ho, C.-Y. Yang, M. I. Jordan. Convergence rates for Gaussian mixtures of experts. Journal of Machine Learning Research, 2022.

[2] H. Hazimeh. Dselect-k: Differentiable selection in the mixture of experts with applications to multi-task learning. In NeurIPS, 2021.

---

### Decision · Program_Chairs · 2023-09-21

**Decision:**

Accept (spotlight)

**Comment:**

The paper studies statistical rates for maximum likelihood estimation in softmax-gated Gaussian mixture of experts, with gating probabilities that are allowed to depend on inputs. Reviewers were happy with the paper's contributions as well as the author response. Some concerns were raised regarding accessibility of presentation and relevance to practice, and I encourage the authors to clarify these points in the final version.